# Pharmacological or genetic inhibition of iNOS prevents cachexia-mediated muscle wasting and its associated metabolism defects

Jason Sadek[1,2,†], Derek T Hall[1,2,3,4,†], Bianca Colalillo[1,2], Amr Omer[1,2], Anne-Marie K Tremblay[1,2], Virginie Sanguin-Gendreau[1,2], William Muller[1,2], Sergio Di Marco[1,2], Marco Emilio Bianchi[5] & Imed-Eddine Gallouzi[1,2,6,*]

## Abstract

Cachexia syndrome develops in patients with diseases such as cancer and sepsis and is characterized by progressive muscle wasting. While iNOS is one of the main effectors of cachexia, its mechanism of action and whether it could be targeted for therapy remains unexplored. Here, we show that iNOS knockout mice and mice treated with the clinically tested iNOS inhibitor GW274150 are protected against muscle wasting in models of both septic and cancer cachexia. We demonstrate that iNOS triggers muscle wasting by disrupting mitochondrial content, morphology, and energy production processes such as the TCA cycle and acylcarnitine transport. Notably, iNOS inhibits oxidative phosphorylation through impairment of complexes II and IV of the electron transport chain and reduces ATP production, leading to energetic stress, activation of AMPK, suppression of mTOR, and, ultimately, muscle atrophy. Importantly, all these effects were reversed by GW274150. Therefore, our data establish how iNOS induces muscle wasting under cachectic conditions and provide a proof of principle for the repurposing of iNOS inhibitors, such as GW274150 for the treatment of cachexia.

**Keywords** cachexia; cancer; inflammation; iNOS; metabolism
**Subject Categories** Metabolism; Musculoskeletal System

## Introduction

Cachexia is a debilitating wasting syndrome that arises in numerous chronic conditions such as cancer, chronic obstructive pulmonary disorder (COPD), chronic heart failure (CHF), HIV infection, and sepsis (Farkas *et al*, 2013). It is characterized by a dramatic, involuntary loss of both lean muscle and adipose tissue mass (Fearon *et al*, 2011; Blum *et al*, 2014). Development of cachexia is associated with a significant increase in morbidity and mortality in patients due to loss of skeletal muscle function (Takayama *et al*, 2016; Naito *et al*, 2017; Vigano *et al*, 2017; Gannavarapu *et al*, 2018; Zhou *et al*, 2018). Despite its significant impact on patient quality of life and chance for survival, there are currently no effective therapies for the treatment of cachexia. It is well accepted that cachexia is triggered by chronic inflammation and the upregulation of several pro-inflammatory cytokines (Fearon *et al*, 2012; Baracos *et al*, 2018). However, monotherapies targeting various cytokines have not successfully treated the condition due to its multifactorial nature (Jatoi *et al*, 2007; Wiedenmann *et al*, 2008; Jatoi *et al*, 2010). Downstream effectors of these pro-inflammatory cytokines within affected tissues, therefore, represent a more viable avenue for the development of effective therapies.

Inflammatory cytokines promote cachexia-induced muscle wasting by triggering a catabolic state in muscle (Hall *et al*, 2011; Argilés *et al*, 2013; Cohen *et al*, 2015). Many cytokines induce the expression of muscle-specific E3 ligases, such as MuRF1 and Atrogin-1/MAFbx, that promote atrophy by targeting protein substrates for degradation (Fearon *et al*, 2012; Baracos *et al*, 2018). Reduced anabolic signaling and protein synthesis have also been demonstrated in murine models of cachexia and human patients (Emery *et al*, 1984; Smith & Tisdale, 1993; White *et al*, 2011; White *et al*, 2013; Puppa *et al*, 2014; Brown *et al*, 2018; Hall *et al*, 2018). Recent observations have suggested that, in addition to alterations in protein homeostasis, cachectic muscle has significant impairment in metabolic functions (Fearon *et al*, 2012; Argiles *et al*, 2015; Porporato, 2016; Hall *et al*, 2018) and metabolome (Der-Torossian *et al*, 2013b; QuanJun *et al*, 2015; Tseng *et al*, 2015b; Cui *et al*, 2019; Kunzke *et al*, 2019; Pin

1 Department of Biochemistry, McGill University, Montreal, QC, Canada
2 Rosalind & Morris Goodman Cancer Research Center, McGill University, Montreal, QC, Canada
3 Sprott Centre for Stem Cell Research, Regenerative Medicine Program, Ottawa Hospital Research Institute, Ottawa, ON, Canada
4 Department of Cellular and Molecular Medicine, Faculty of Medicine, University of Ottawa, Ottawa, ON, Canada
5 Division of Genetics and Cell Biology, Chromatin Dynamics Unit, IRCCS San Raffaele Scientific Institute and Vita-Salute San Raffaele University, Milan, Italy
6 KAUST Smart-Health Initiative and Biological and Environmental Science and Engineering (BESE) Division, King Abdullah University of Science and Technology (KAUST), Jeddah, Saudi Arabia
*Corresponding author. Tel: +1 514 398 4537; E-mails: imed.gallouzi@mcgill.ca; gallouzi.imed@kaust.edu.sa
†These authors contributed equally to this work

et al, 2019a; Lautaoja et al, 2019b). The mitochondrial dysfunction observed under these cachectic conditions is likely due to impaired oxidative phosphorylation (OXPHOS) that is associated with reduced electron transport chain (ETC) complex activity (Julienne et al, 2012; Puppa et al, 2012; Der-Torossian et al, 2013b; Fermoselle et al, 2013; Padrao et al, 2013; McLean et al, 2014; Brown et al, 2017; Hall et al, 2018; VanderVeen et al, 2018; Pin et al, 2019a). In addition, other mitochondrial defects, such as increased uncoupling of the proton gradient from ATP production, altered fission, and fusion, and abnormal morphology have been described (Shum et al, 2012; White et al, 2012; Fontes-Oliveira et al, 2013; Tzika et al, 2013; Antunes et al, 2014). This loss of mitochondrial function is correlated with a reduction in the rate of ATP production, suggesting that the energy balance of muscle is negatively impacted (Constantinou et al, 2011; Tzika et al, 2013; Antunes et al, 2014; Pin et al, 2019a). Many studies investigating metabolic impairment in cachectic muscle have demonstrated that these changes are due to the activation of the NF-κB pathway by pro-inflammatory cytokines such as TNF-α and IL-6 (White et al, 2012; Der-Torossian et al, 2013b; Fermoselle et al, 2013), and the NF-κB-dependent mechanisms that drive cytokine-mediated mitochondrial dysfunction in cachexia, however, have yet to be delineated.

Inducible nitric oxide synthase (iNOS; NOS2), which is highly expressed in cachectic muscle, is a known downstream effector of the NF-kB pathway (Buck & Chojkier, 1996; Di Marco et al, 2005; Ramamoorthy et al, 2009; Hall et al, 2011; Di Marco et al, 2012; Ma et al, 2017; Hall et al, 2018). Nitric oxide synthases (NOS) produce nitric oxide (NO) from the breakdown of L-arginine to L-citrulline (Strijdom et al, 2009). iNOS is one of three isoforms of NOS, together with endothelial (eNOS) and neuronal (nNOS). While eNOS and nNOS are associated with physiological NO production and cellular signaling, iNOS expression and activity are strongly induced by inflammation. Although the iNOS/NO pathway has been previously linked to the onset of cachexia-induced muscle loss (Buck & Chojkier, 1996; Barreiro et al, 2005; Di Marco et al, 2005; Szabo et al, 2007; Hall et al, 2011; Di Marco et al, 2012; Wiseman & Thurmond, 2012; Ma et al, 2017; Hall et al, 2018), the mechanisms whereby excessive NO contributes to muscle wasting are still poorly understood.

In this study, we demonstrate that the iNOS/NO pathway is a trigger of the metabolome derangement, mitochondrial dysfunction, and energy crisis associated with the onset of inflammation-induced muscle wasting. Specifically, iNOS mediates these effects by impairing mitochondrial metabolic processes such as the TCA cycle, the ETC, and acylcarnitine metabolism in skeletal muscle. Importantly, our data demonstrate that interfering with iNOS function pharmacologically or via a genetic depletion prevents all the metabolic defects described above as well as muscle wasting. We suggest that iNOS inhibition is a viable therapeutic strategy to prevent cachexia-induced metabolic derangement and, ultimately, muscle loss.

# Results

## Genetic deletion of the iNOS gene prevents inflammation-induced energy crisis and muscle wasting

An accumulating body of evidence points to iNOS as one of the main drivers of inflammation and cancer-induced muscle wasting

(Di Marco et al, 2005; Der-Torossian et al, 2013b; Fermoselle et al, 2013; He et al, 2013). Since pathological production of NO by iNOS has been linked to metabolic dysfunction in inflammatory diseases (Anavi & Tirosh, 2019), we tested whether this can also be the case in mouse models of inflammation-induced muscle loss.

We assessed muscle loss and the associated metabolic response in wild-type (WT) and iNOS knockout (KO) mice in a lipopolysaccharide (LPS)-induced model of septic cachexia. Male wild-type (WT) and iNOS KO C57BL/6 mice were injected or not with a dose of LPS previously shown to quickly induce muscle wasting over an 18-hour period (Jin & Li, 2007; Braun et al, 2013; Hall et al, 2018). First, we assessed iNOS expression and activity in the muscles of WT and iNOS knockout mice treated with or without LPS. We confirmed, as expected, that the WT mice, but not the iNOS KO mice, expressed iNOS protein in response to LPS (Fig 1A). The activation of the iNOS/NO pathway in these muscles was determined by assessing the formation of 3-nitrotyrosine (3NT)-modified proteins (Chatterjee et al, 2003; Nanetti et al, 2007; Ahsan, 2013; Wei et al, 2015). We demonstrated that the levels of 3-NT-modified proteins were only increased in LPS-treated WT but not iNOS KO mice (Fig 1B).

We then assessed activation of the immune response in these two mouse strains treated with or without LPS for 18 h, with particular attention to macrophage polarization and levels due to their prominent role in the LPS response (Beutler & Rietschel, 2003; Ginhoux et al, 2016). We observed that LPS had similar effects on the mass of the spleen (Fig EV1A), the population of splenic M1/M2 macrophages (Fig EV1B–D), and the accumulation of macrophages in the muscle (Fig EV1E) of WT and iNOS KO mice. Despite this, the LPS-induced systemic levels of the pro-cachectic cytokines IL-6 and TNF-α (but not IL-1β, and IL-1α) were significantly lower in iNOS KO mice compared with their WT counterparts (Dataset EV1 and Fig EV1F–I). Our results demonstrate that iNOS affects the immune response by mediating the secretion of systemic pro-inflammatory cytokines such as IL-6 and TNF-α without affecting the polarization of macrophages.

Next, we evaluated whether the genetic ablation of iNOS affects LPS-induced changes in body weight and muscle mass. As previously described, the mass of skeletal muscle and tissues was normalized to the initial body weight (Michaelis et al, 2017; Parajuli et al, 2018; Pin et al, 2019a; Pin et al, 2019b). LPS, as previously shown, is known to affect food consumption rates in mice (Braun et al, 2013; Hall et al, 2018). Therefore, the saline-treated WT as well as the saline and LPS-treated iNOS KO mice were pair-fed to match the consumption rate of the LPS-treated WT mice in order to account for variations which may occur due to differences in food consumption. We observed that although pair feeding caused a 10–12% decrease in body weight in both saline-treated WT and iNOS KO mice, treatment with LPS did not further exacerbate this loss of mass (Fig EV2A). The genetic ablation of iNOS, however, did prevent the LPS-induced wasting of the tibialis anterior (TA), soleus, and quadriceps muscles but not the gastrocnemius muscle (Fig 1C–F and EV2B). The iNOS-induced loss of muscle mass was further confirmed by assessing the minimum Feret diameter (Fig 1G) and cross-sectional area (CSA) (Fig EV2C) of the muscle fibers. It is well-established that inflammation-induced muscle wasting is often associated with a functional deficit in muscle strength (Murphy et al, 2012). By measuring grip strength, we found that the iNOS KO

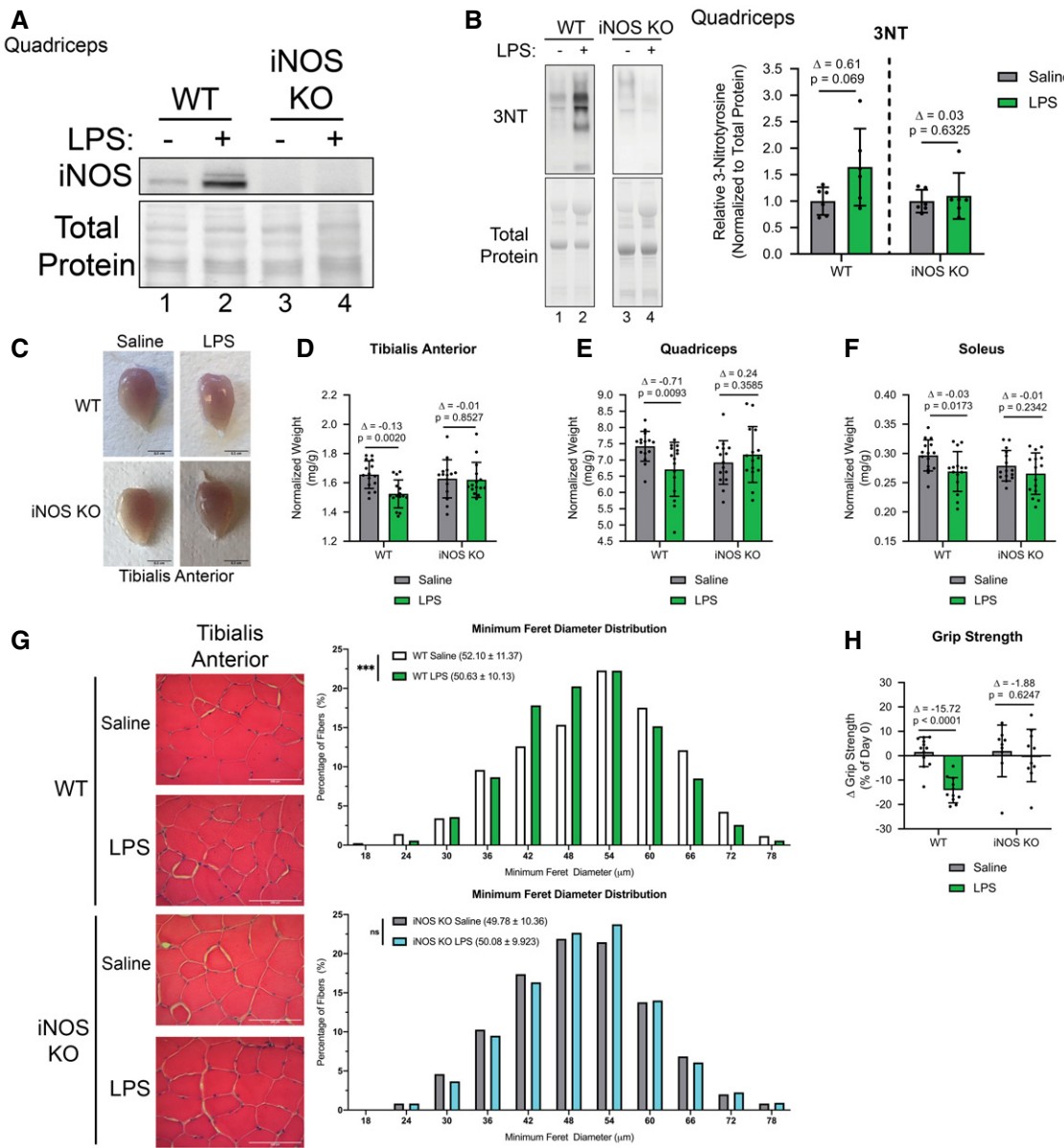

**Figure 1.  iNOS knockout mice are resistant to LPS-driven muscle wasting.**

Male C57BL/6 wild-type (WT) and iNOS knockout (KO) mice were intraperitoneally injected with 1 mg kg$^{-1}$ LPS or an equivalent volume of carrier solution. Control WT, control KO, and LPS-treated KO cohorts were pair-fed (PF) to the WT LPS-treated cohorts. After 18 h, mice were euthanized, and tissue samples were analyzed.

A       *Quadriceps* from saline or LPS-treated, WT, or iNOS KO mice were isolated and used for Western blot analysis with anti-iNOS ($n = 4$). Ponceau S was used for total protein staining.

B       *(left)* Quadriceps from saline or LPS-treated, WT, or iNOS KO mice were isolated and used for Western blot with anti-3NT ($n = 6$). Total protein levels are also shown. *(right)* Quantification of the 3NT-to-total protein ratio. Ratios are expressed relative to the saline-treated controls.

C       Representative images of tibialis anterior muscles. Scale bars represent 0.5 cm.

D–F    Weight of *tibialis anterior* (D), *quadricep* (E), and *soleus* (F) muscle normalized to initial body weight ($n = 15$).

G       *(left)* Representative photomicrographs of H&E-stained *tibialis anterior* muscle sections from control and iNOS KO mice injected with or without LPS. Scale bars = 100 μm. *(right)* Frequency histogram showing the distribution of muscle fiber minimum Feret diameter in the *tibialis anterior* muscles from saline- or LPS-treated *(top)* WT control and *(bottom)* iNOS KO mice ($n = 4$). A total of 300 fibers per muscle were used for the analysis. Statistical comparisons, mean, and standard deviation of the mean are shown in the histogram legend.

H       Change in grip strength between day of injection and endpoint of experiment. (WT saline $n = 12$, WT LPS $n = 11$, iNOS KO saline $n = 9$, and iNOS KO LPS $n = 10$).

Data information: Individual data points represent values from individual mice. Error bars represent the standard deviation (SD) of the mean. (B) Statistical comparisons were made between saline-treated controls and LPS-treated mice of same genotype. Plotted data were relativized to saline controls of corresponding genotype. Δ indicates the difference in mean values, and *P*-values were calculated with Student's *t*-test. (D–F and H) For statistical comparisons, Δ indicates the difference in mean values and *P*-values were calculated with an ANOVA followed by Fisher's LSD test. (G) *P*-values were calculated with a Kolmogorov–Smirnov test (***$P < 0.001$). Non-statistically significant comparisons ($P > 0.05$) are indicated as non-significant (ns).

Source data are available online for this figure.

mice, but not their WT counterparts, were completely protected from LPS-mediated loss of muscle strength (Fig 1H). These observations clearly show that genetic disruption of iNOS protects mice from sepsis-induced muscle wasting.

To determine whether the impact of iNOS on muscle wasting is mediated by changes in muscle metabolism, we next performed LC-MS/MS to determine the levels of 119 key metabolites including biogenic amines, amino acids, acylcarnitines, glycerophospholipids, and organic acids (Dataset EV2) in *tibialis anterior* (TA) muscles collected from the WT and iNOS KO mice treated or not with LPS. The raw metabolite concentration data obtained were analyzed globally using the MetaboAnalyst 4.0 software and were mean-centered and range-scaled to allow for comparison of metabolites within their biological range (Chong *et al*, 2018; Chong *et al*, 2019a; Chong & Xia, 2018; Chong *et al*, 2019b; van den Berg *et al*, 2006; Xia *et al*, 2012; Xia *et al*, 2009; Xia *et al*, 2015; Xia & Wishart, 2011a, b, 2016). We observed, through the multivariate approach of partial least squares discriminant analysis (PLS-DA), that the metabolomic profile of the TAs from WT mice is distinct from their iNOS KO counterparts and that LPS causes metabolomic shifts in both mouse backgrounds (Fig 2A). Analysis of heatmaps comparing the WT saline cohort to the iNOS KO saline cohort and the WT LPS cohort to the iNOS KO LPS cohort further confirmed the presence of the distinct metabolic profiles (Appendix Fig S1A–C). To assess whether differences in the observed distinct metabolic profiles found in our four treatment groups were correlated with differences in muscle fiber type composition, we assessed the distribution of type I, type IIA, and type IIB/IIX fibers in the TAs of these mice. We observed no appreciable differences between the four groups, indicating that other factors were responsible for the metabolic differences between them (Fig EV2D). Pathway analysis using the KEGG library in *Mus musculus* identified several pathways, including arginine metabolism (as expected), glycolysis, and tricarboxylic acid (TCA) cycle, that were altered in WT mice. These pathways, however, were not affected in the iNOS KO mice (Fig 2B). The impairment of several pathways involving energy production suggests that iNOS triggers an energetic crisis in wasting muscles. An increase in the AMP/ATP ratio, due to energetic stress, is known to activate AMPK (5'-activated protein kinase), an important regulator of energy homeostasis, through phosphorylation at Thr172 (pAMPK; Stein *et al*, 2000; Hardie *et al*, 2012). Western blot analysis of the *quadriceps* muscle showed that LPS activates AMPK in WT but not iNOS KO mice (Fig 2C). Collectively, these data strongly suggest that LPS-induced iNOS expression, in skeletal muscle, disrupts normal energy metabolism prompting energetic stress.

Next, we sought to determine the mechanism by which iNOS impairs cellular energy production in skeletal muscles undergoing wasting. We examined the concentrations of specific metabolites in key pathways involved in cellular energy production, including glycolysis, the TCA cycle, and acylcarnitine metabolism (Fig 2D and G). Analysis of heatmaps revealed that most of the detected metabolites involved in glycolysis and the TCA cycle are altered by LPS treatment of WT but not the iNOS KO mice (Appendix Fig S2A and B). Within glycolysis, we observed that iNOS induced a trend toward an increase in glucose levels as well as a trend toward a decrease in pyruvate levels, suggesting a potential reduction in glucose utilization in WT LPS-treated mice. We additionally observed a dramatic iNOS-dependent decrease in the TCA cycle

intermediate α-ketoglutarate (α-KG). We also detected a trend toward a decrease in succinate levels, as well as an increase in citrate (Fig 2D–E). Taken together with our pathway analysis, these iNOS-dependent alterations to TCA cycle intermediates also suggest an involvement of iNOS in glutamine metabolism during LPS-induced muscle wasting (Fig 2B and D–E). The levels of arginine, which is known to regulate the TCA cycle and oxidative metabolism (El-Gayar *et al*, 2003; Lee *et al*, 2003; Chaturvedi *et al*, 2007; Xu *et al*, 2016; Kunzke *et al*, 2019), were also significantly increased in response to LPS in an iNOS-dependent manner (Fig 2F). We then investigated β-oxidation, which is used to produce energy by several tissues, including muscle (Henique *et al*, 2015). Carnitine palmitoyl-transferase 1 (CPT1) and CPT2 catalyze the rate-limiting steps of β-oxidation. Acyl-CoA molecules are converted to acylcarnitine species by CPT1 to allow for their transport into the mitochondria, where acylcarnitines are converted back to their CoA equivalents by CPT2 (Zhou *et al*, 2012; Wu *et al*, 2017; Li *et al*, 2019). Once converted back to their CoA analog, fatty acids enter β-oxidation through long-chain Acyl-CoAs dehydrogenase (LCADH; Zhou *et al*, 2012; Wu *et al*, 2017; Li *et al*, 2019) (Fig 2G). We observed a global decrease in long, even-chained acylcarnitine species in LPS-treated WT mice, but not in iNOS KO mice (Appendix Fig S3). We next estimated the activities of CPT1, CPT2, and LCADH as well as β-oxidation by calculating the ratios of various acylcarnitine combinations (Zhou *et al*, 2012; Wu *et al*, 2017; Li *et al*, 2019). We observed that in response to LPS treatment, the global activity of CPT1 was decreased in an iNOS-dependent manner (Fig 2H). CPT2 activity also tended to decrease (Fig 2H), suggesting that during cachexia, iNOS impairs the activities of CPT1 and CPT2.

Together, our data suggest that LPS treatment interferes with cellular energy production and mitochondrial function in an iNOS-dependent manner. iNOS mediates these effects by impairing glycolysis and disrupting the entry of both pyruvate (via its decrease) and fatty acids (via inhibition of CPT1 and CPT2) into the TCA cycle.

### The iNOS inhibitor GW274150 prevents cancer-induced muscle wasting

The data outlined above raise the possibility that targeting iNOS could represent a viable option to prevent inflammation-induced muscle wasting. To provide a proof of principle for this possibility, we looked for an iNOS inhibitor that is efficient in an *in vivo* setting without having major side effects. GW274150 (GW) has been established as a highly specific inhibitor for iNOS (over eNOS and nNOS) and has been tested in phase II clinical trials to treat migraines, rheumatoid arthritis, and asthma. Although GW was found to be safe, its effectiveness was not better than pre-existing drugs against these conditions (Høivik *et al*, 2010; Seymour *et al*, 2012; Vitecek *et al*, 2012).

Therefore, we decided to investigate the possibility of repurposing GW to treat muscle wasting. We first used our sepsis mouse model to test whether GW could mimic the effects of iNOS depletion in preventing LPS-induced muscle wasting. GW was administered at a dose of 5 mg kg$^{-1}$, which was previously shown to reduce collagen-induced arthritis symptoms in mice to an extent similar to that in iNOS knockout mice (Cuzzocrea *et al*, 2002). When compared to the results obtained in Figs 1D and H, Fig EV1A, and Fig EV2A on WT and iNOS KO mice treated with or without LPS,

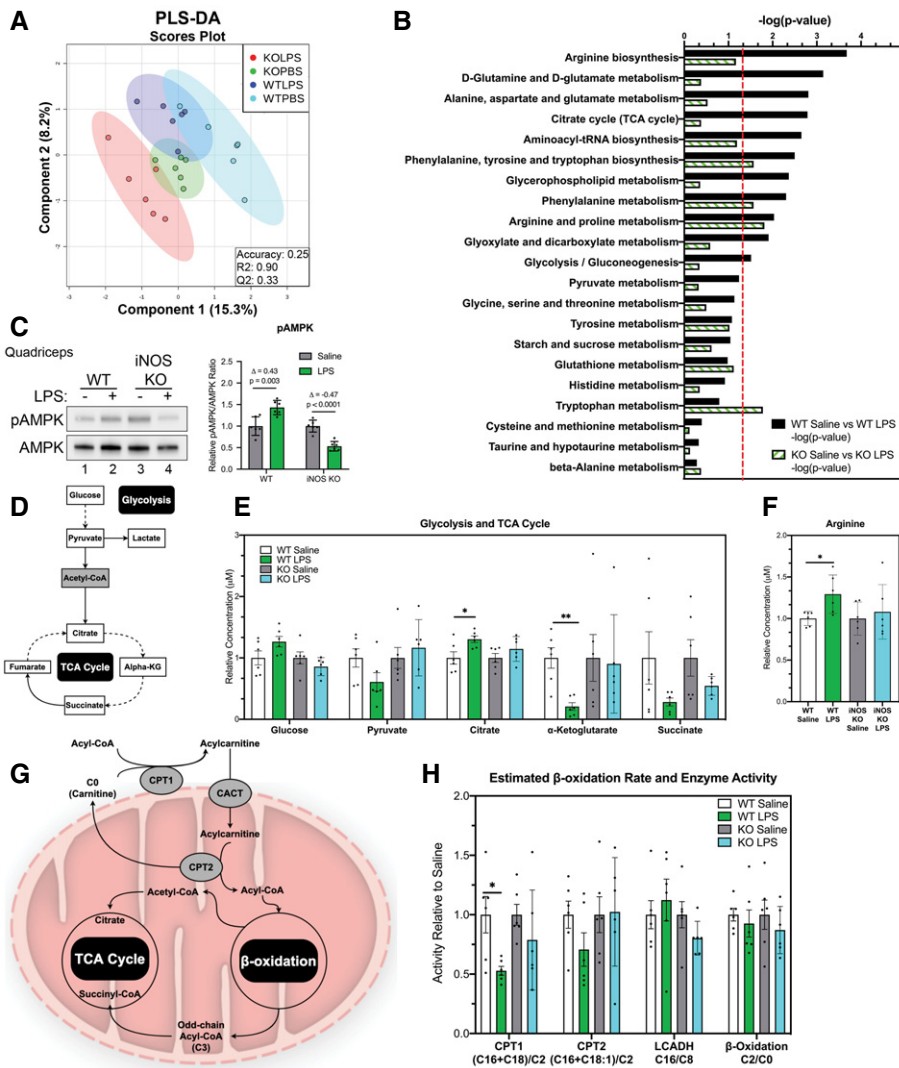

**Figure 2. Genetic ablation of iNOS prevents LPS-driven deregulation of the TCA cycle and energy production.**

Male C57BL/6 wild-type (WT) and iNOS knockout (KO) mice were intraperitoneally injected with 1 mg kg$^{-1}$ LPS or an equivalent volume of carrier solution. Control WT, control KO, and LPS-treated KO cohorts were pair-fed (PF) to the WT LPS-treated cohorts. After 18 h, mice were euthanized and the metabolome of tibialis anterior muscles was analyzed through LC-MS/MS.

A   Scores scatter plot of partial least square discriminant analysis (PLS-DA) model of metabolites from WT and iNOS KO mice treated with or without LPS. Metabolomic data were range-scaled and mean-centered.

B   Pathway analysis using MetaboAnalyst 4.0 Software comparing significantly altered pathways from WT saline to LPS as well as iNOS KO saline to LPS. Pathways are ranked by their significance and filtered based on a Pathway Impact Score > 0.1. Metabolomic data were range-scaled and mean-centered. *P*-values were obtained using GlobalTest, and the −log(*P*-value) corresponding to a *P*-value of 0.05 is indicated by the red dashed line.

C   (*left*) Western blot analysis of pThr172-AMPK (pAMPK) and total AMPK (AMPK) in *quadriceps* muscle. (*right*) Quantification of the pAMPK-to-AMPK ratio. Ratios are expressed relative to the saline-treated controls.

D   Schematic of glycolysis and the TCA cycle. White boxes denote detected metabolites. Gray boxes denote undetected metabolites. Dashed arrows indicate the presence of multiple reactions between metabolites.

E   Relative concentrations of metabolites involved in glycolysis and TCA cycle.

F   Relative concentration of arginine.

G   Schematic of acylcarnitine metabolism. Gray circles denote key enzymes involved in metabolic processes highlighted.

H   Relative estimated activity of CPT1, CPT2, LCADH, and β-oxidation.

Data information: Individual data points represent values from individual mice, with a total of six mice per cohort (*n* = 6). (B) *P*-values were obtained using GlobalTest. (C–H) Error bars represent the standard deviation (SD) of the mean. Statistical comparisons were made between saline-treated controls and LPS-treated mice of same genotype. Plotted concentration data were relativized to saline controls of corresponding genotype. *P*-values were calculated with Student's *t*-test (**P* < 0.05; ***P* < 0.01). (C) Δ indicates the difference in mean values.

Source data are available online for this figure.

we demonstrated that this dose of GW, similar to that was observed in iNOS KO, protected mice against LPS-induced muscle wasting (Appendix Fig S4A–D).

Next, we assessed the efficacy of GW-mediated inhibition of iNOS in the C26 adenocarcinoma mouse model of cancer cachexia (Di Marco *et al*, 2012; Murphy *et al*, 2012; Der-Torossian *et al*, 2013b; Hall *et al*, 2018; Lautaoja *et al*, 2019a; Pin *et al*, 2019a; Pin *et al*, 2019b). For these experiments, we injected male BALB/C mice with C26 colon cancer cells, a treatment known to induce muscle wasting over the course of 14–19 days (Di Marco *et al*, 2012; Bonetto *et al*, 2016; Hall *et al*, 2018). First, we confirmed that iNOS expression was induced in the muscle of C26-bearing mice (Fig EV3A), as we have previously shown (Di Marco *et al*, 2012). Since iNOS inhibition has been reported to impair tumor growth (Kostourou *et al*, 2011; Granados-Principal *et al*, 2015; Garrido *et al*, 2017), mice were treated with GW 5 days post-injection of the C26. We demonstrated that GW, under these conditions, did not have any effect on tumor burden (Fig 3A). To assess the effectiveness of GW in impairing the iNOS/NO pathway in skeletal muscle, we next measured levels of 3NT-modified protein as described above (Chatterjee *et al*, 2003; Nanetti *et al*, 2007; Ahsan, 2013; Wei *et al*, 2015). We observed a C26-mediated increase in 3NT-modified protein levels that were significantly decreased by GW treatment, demonstrating that GW impaired iNOS activity in muscle (Fig 3B). We then characterized the immune response of C26 tumor-bearing mice treated with or without GW by assessing spleen mass, muscle macrophage content (assessed by verifying F4/80 expression), and serum cytokine levels (Dataset EV3). We observed that although spleen mass and muscle macrophage content increased in C26-tumor-bearing mice when compared to their saline-control counterparts, these changes were not affected by treatment with GW (Fig EV3B and C). The C26 model of cancer-induced muscle wasting is known to be mediated by elevated levels of pro-inflammatory cytokines such as IL-6 (Narsale & Carson, 2014). Interestingly, we observed that although the serum levels of several cytokines including IL-6, TNF-α, and IL-1β (but not IL-1α) increased in C26-tumor-bearing mice, these effects were prevented by treatment with GW (Fig EV3D–G). Similar to our observations in the septic model of cachexia, the above data suggest that GW affects the C26-mediated immune response by preventing the secretion of these factors by inflammatory cells.

Next, we evaluated the effect of GW on C26-induced cachexia. In both cohorts (with and without GW), we observed significant body weight loss, although less severely in the GW-treated mice (Fig 3C). GW treatment did not affect adipose tissue wasting, which likely contributed to the observed overall body weight change (Fig EV3H). However, GW treatment partially protected against the C26-mediated atrophy of the TA, *soleus*, *quadriceps*, and *gastrocnemius* muscles (Fig 3D–G). These data were corroborated through analyses of the minimum ferret diameter and CSA of *gastrocnemius* muscle fibers, which demonstrated a protective effect of GW against C26-induced atrophy (Figs 3H and EV3I). We observed a significant decrease in grip strength in C26-treated mice (Fig 3I). Although GW treatment alone tended to decrease grip strength, it did partially rescue (albeit non-significantly) the loss of strength observed in C26-treated mice (Fig 3I). Therefore, these observations suggest that GW-mediated inhibition of iNOS protects mice from cancer-induced muscle atrophy and loss of muscle function.

## The iNOS inhibitor GW274150 prevents deleterious effects on energy production observed during cancer-induced muscle wasting

Given the protective effect of GW against C26-induced muscle wasting and the reported metabolomic derangement of skeletal muscle of C26-treated mice (Murphy *et al*, 2012; Der-Torossian *et al*, 2013b; Lautaoja *et al*, 2019a; Pin *et al*, 2019a; Pin *et al*, 2019b), we next assessed the impact of GW and its inhibitory effect on iNOS function on the skeletal muscle metabolome in our model. LC-MS/MS metabolomic analysis on TA muscles (Dataset EV4) showed that GW alone had minimal effects on the skeletal muscle metabolome, only affecting arginine biosynthesis and D-glutamine/D-glutamate metabolism (Appendix Fig S5A–C). We next used the same metabolomic approach to compare metabolite profiles on TA samples collected from control or C26-treated mice treated with or without GW (Dataset EV5). PLS-DA analysis showed global cancer-induced shifts in the overall metabolome (Fig 4A). Pathway analysis showed that, in skeletal muscle undergoing wasting, GW targeted a few key pathways that were affected by C26 tumors (Fig 4B). Similar to our observations in the LPS model, these pathways include those associated with energy production (glycolysis and pyruvate metabolism) and amino acid metabolism (Fig 4B and Appendix Fig S6A and B). Immunofluorescence analysis of the expression levels of myosin heavy chains I and IIa as well as laminin indicated that these effects on the metabolic profiles were not due to a shift in the distribution of muscle (type I, type IIA, and type IIB/IIX) fiber types in the *gastrocnemius* muscles of the treated mice (Fig EV3J). Next, we evaluated the impact of iNOS on the global energetic status of these mice by measuring AMPK activation. In line with our observations in the sepsis model, although AMPK was activated in muscle of C26-tumor-bearing mice, this effect was partially prevented by GW (Fig 4C). GW treatment alone, however, did not have any effect on AMPK activation (Fig 4C). Therefore, our data suggest that iNOS activity disrupts normal energy metabolism in skeletal muscle both in a sepsis and in a cancer-induced cachexia model.

Next, we determined how iNOS promotes the energy crisis in cancer cachexia. Although the C26 tumor did not affect the relative concentration of glucose, GW significantly decreased glucose levels in C26-tumor muscles (Fig 4D). In addition, GW partially prevented an increase in arginine levels and other iNOS-dependent amino acid derangements in C26-tumor mice (Fig 4E). In particular, lysine, methylhistidine, and tryptophan levels were increased, and aspartate trended toward a decrease in C26-tumor mice, effects which were recovered due to GW treatment. These changes in amino acid levels in muscle are consistent with a number of recent studies assessing the metabolome of cancer cachectic muscle (Der-Torossian *et al*, 2013a; QuanJun *et al*, 2015; Tseng *et al*, 2015a; Kunzke *et al*, 2019; Lautaoja *et al*, 2019b). It has been suggested that elevation of methylhistidine in cachectic conditions is associated with muscle protein breakdown (QuanJun *et al*, 2015). Therefore, the recovery of normal methylhistidine levels in C26-tumor-bearing mice treated with GW could indicate a protective effect of iNOS inhibition against protein catabolism. Interestingly, increased lysine and arginine levels in cachectic muscle have been linked to defects in energy production and energetic stress (Kunzke *et al*, 2019). The recovery of these amino acid levels in GW-treated mice indicates, therefore, a possible protection against energetic crisis in our model.

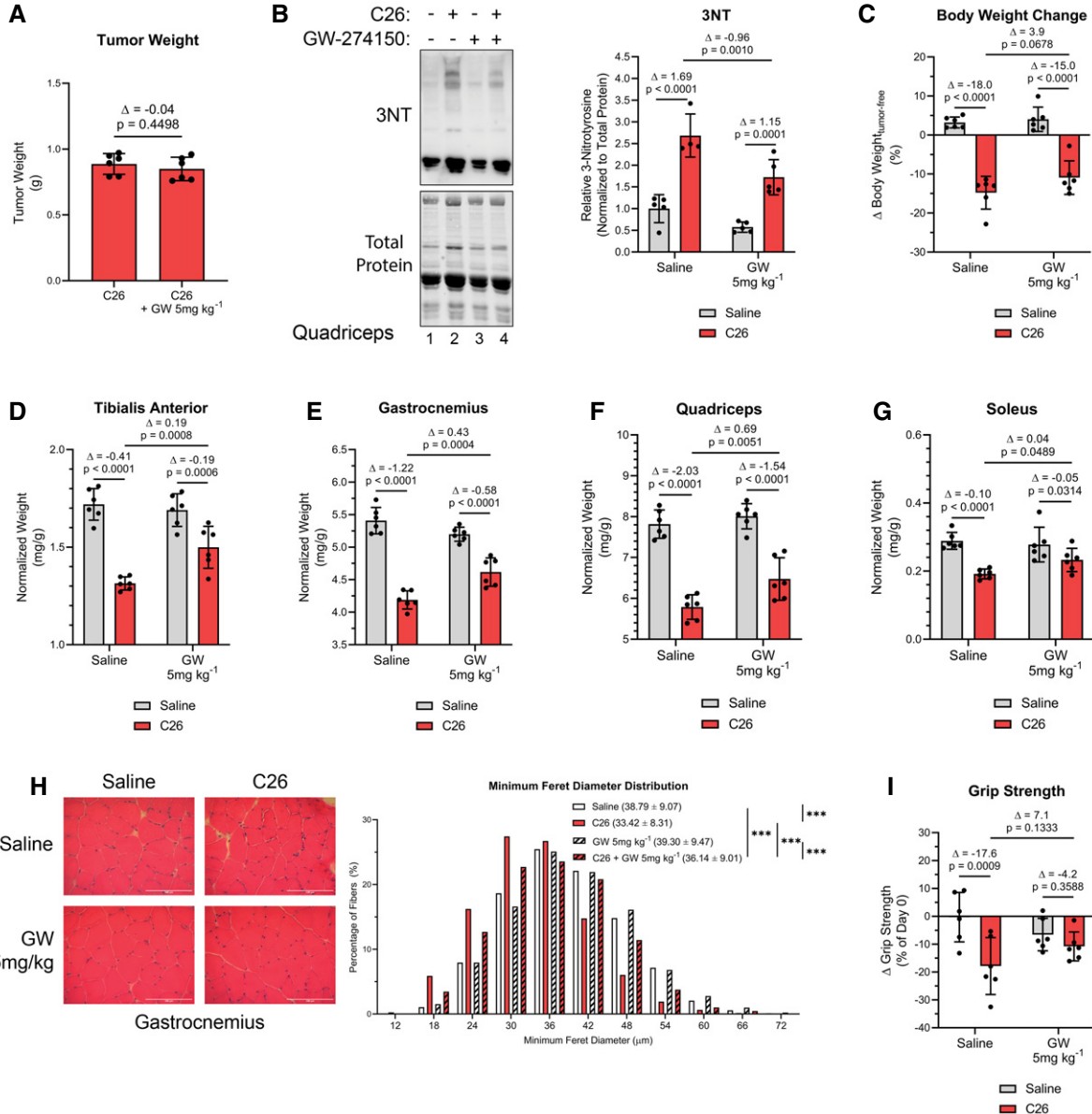

**Figure 3. GW274150 treatment reduces muscle wasting in the C26 model.**

Male BALB/C mice were injected subcutaneously with C26 cells ($1.25 \times 10^6$ cells) or an equivalent volume of saline. After 5 days, and everyday thereafter, saline and C26 injected mice were treated with or without GW (5 mg kg$^{-1}$). After 16 days, mice were euthanized and tissue samples were analyzed.

A　　Effect of GW on tumor weight.

B　　(*left*) Representative image of Western blot analysis of 3NT staining and stain free imaged total protein levels from *quadriceps* muscle. (*right*) Quantification of the 3NT-to-total protein ratio (saline *n* = 5, C26 *n* = 4, GW 5 mg/kg *n* = 5, and C26 GW 5 mg/kg *n* = 5).

C　　Percent body weight change from day 0 to day 16.

D–G　Weight of *tibialis anterior* (D), *gastrocnemius* (E), *quadricep* (F), and *soleus* (G) muscle normalized to initial body weight.

H　　(*left*) Representative photomicrographs of *gastrocnemius* muscle sections from control and C26 mice treated with or without GW taken after H&E staining. Scale bars = 100 μm. (*right*) Frequency histogram showing the distribution of muscle fiber minimum Feret diameter in the *gastrocnemius* muscles from control and C26 mice treated with or without GW (saline *n* = 4, C26 *n* = 4, GW 5 mg/kg *n* = 3, and C26 GW 5 mg/kg *n* = 4). A total of 600-700 fibers per muscle were used for the analysis. Statistical comparisons, mean, and standard deviation of the mean are shown in the histogram legend.

I　　Change in grip strength from before tumor cell injection (day 0) and before endpoint collection (day 16).

Data information: Individual data points represent values from individual mice, with a total of six mice per cohort (*n* = 6) unless stated otherwise. Error bars represent the standard deviation (SD) of the mean. (A) Δ indicates the difference in mean values, and *P*-values were calculated with Student's *t*-test. (B–G, I) For statistical comparisons, Δ indicates the difference in mean values and *P*-values were calculated with an ANOVA followed by Fisher's LSD test. (H) *P*-values were calculated with a Kolmogorov–Smirnov test (***$P < 0.001$).

Source data are available online for this figure.

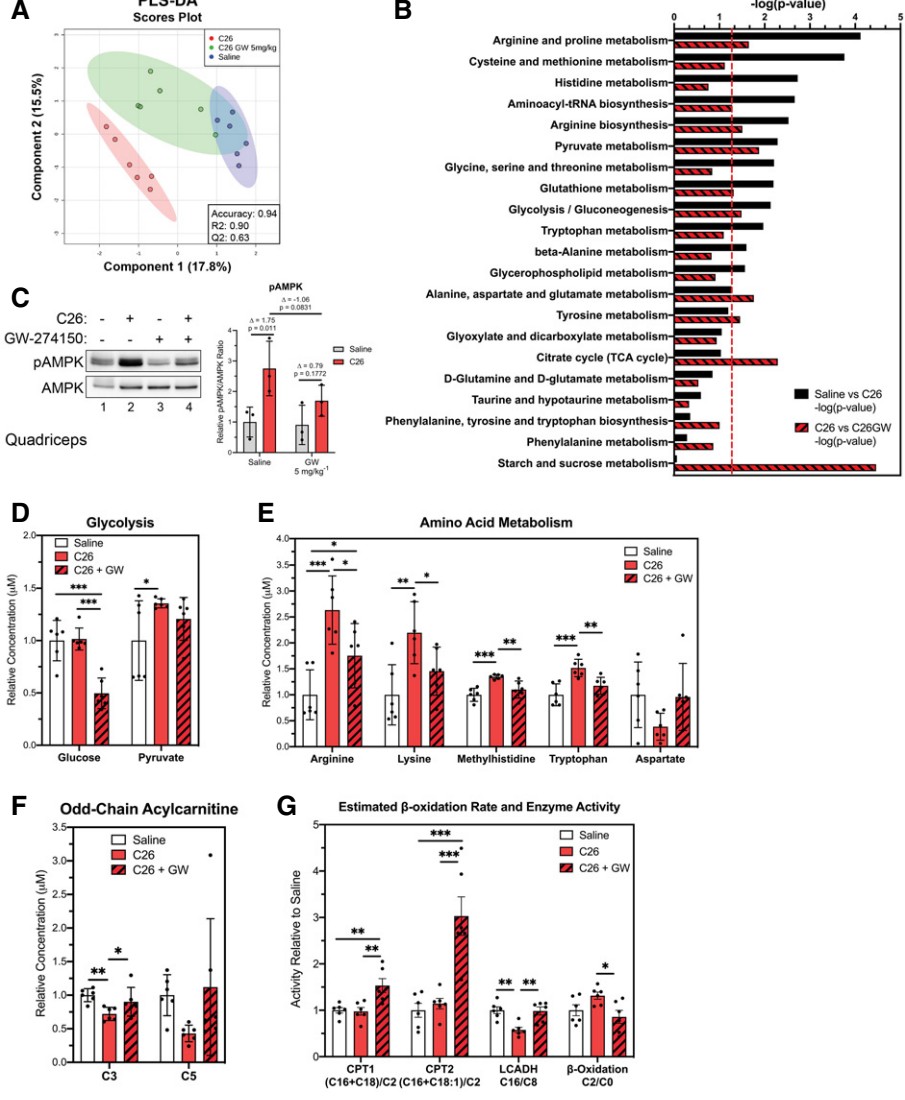

**Figure 4. Pharmacological inhibition of iNOS reduces C26-induced derangement of amino acids and impairment of energy production.**

Male BALB/C mice were injected subcutaneously with C26 cells ($1.25 \times 10^6$ cells) or an equivalent volume of saline. After 5 days and everyday thereafter, the C26 tumor-bearing mice were injected with either saline or GW 5 mg kg$^{-1}$. After 16 days, mice were euthanized and the metabolome of tibialis anterior muscles was analyzed through LC-MS/MS.

A   Scores scatter plot of partial least square discriminant analysis (PLS-DA) model of metabolites from saline, C26, and C26 + GW-treated mice. Metabolomic data were range-scaled and mean-centered.

B   Pathway analysis using MetaboAnalyst 4.0 Software comparing significantly altered pathways from saline to C26 as well as C26 to C26 + GW. Pathways are ranked by their significance and filtered based on a Pathway Impact Score > 0.1. Metabolomic data were range-scaled and mean-centered. P-values were obtained using GlobalTest, and the $-$log(P-value) corresponding to a P-value of 0.05 is indicated by the red dashed line.

C   (left) Western blot analysis of pThr172-AMPK (pAMPK) and total AMPK (AMPK) in quadriceps muscle (n = 3). (right) Quantification of the pAMPK-to-AMPK ratio relative to the saline-treated control (n = 3).

D   Relative concentrations of metabolites involved in glycolysis.

E   Relative concentrations of amino acids.

F   Relative concentrations of odd-chain acylcarnitines.

G   Relative estimated activity of CPT1, CPT2, LCADH, and β-oxidation.

Data information: Individual data points represent values from individual mice, with a total of six mice per cohort (n = 6) unless otherwise stated. (B) P-values were obtained using GlobalTest. (C–G) Error bars represent the standard deviation (SD) of the mean. P-values were calculated with an ANOVA followed by Fisher's LSD test (*P < 0.05; **P < 0.01; ***P < 0.001). (C) Δ indicates the difference in mean values.

Source data are available online for this figure.

As acylcarnitine metabolism was significantly affected by iNOS in our LPS model, we assessed whether this is also the case in our C26 model. We observed a decrease in saturated, long, even-chain acylcarnitines in C26 tumor mice, similar to WT mice treated with LPS (Appendix Fig S7A and B). Interestingly, GW treatment either recovered normal levels or increased the concentration of many long-chain acylcarnitines (e.g., C16 and C18) in C26 tumor-bearing mice (Appendix Fig S7A and B). Mice treated with GW alone also exhibited increases in many long-chain acylcarnitines, indicating that the GW-specific effect occurs independently of the tumor environment (Appendix Fig S8A). Short, odd-chain acylcarnitines were also decreased in cancer cachectic mice and recovered with GW treatment (Fig 4F). It is well-established that these acylcarnitine chains act as anapleurotic metabolites, which feed into the TCA cycle through conversion into succinyl-CoA (Brunengraber & Roe, 2006; Pfeuffer & Jaudszus, 2016). As GW widely affected the acylcarnitine profile of C26 tumor mice, we determined the activity levels of CPT1, CPT2, LCADH, and β-oxidation. Although activity of CPT1 and CPT2 was not significantly affected between control and C26 tumor mice, we observed an increase in their activity in the C26 + GW cohort (Fig 4G). In addition, LCADH and β-oxidation appeared to have iNOS-dependent alterations to their activity, as C26 tumors decreased LCADH activity and increased β-oxidation, both of which were prevented by GW (Fig 4G). Of note, treatment of mice with GW alone affected CPT1, CPT2, and LCADH activity and β-oxidation similarly to what was observed when used to treat C26 tumor-bearing mice (Appendix Fig S8B). Our data therefore indicate that iNOS, in the C26-adenocarcinoma model, affects acylcarnitine-mediated energy production in the mitochondria by interfering with the relationship between acylcarnitine metabolism and the TCA cycle as well as deregulating critical enzymes in β-oxidation.

### iNOS activity affects the wasting of myotubes treated with pro-cachectic cytokines

The above metabolomic analysis demonstrated that iNOS drives muscle wasting by impairing cellular metabolism that leads to an energetic crisis. It is well-established that the induction of energy crisis is directly related to ATP levels, which are controlled by rates of glycolysis and oxidative phosphorylation, ultimately dictated by the enzymes carrying out these processes (Stein et al, 2000; Hardie et al, 2012; Mookerjee et al, 2017, 2018). Therefore, we wanted to understand whether and how iNOS affect ATP production rates from glycolysis and oxidative phosphorylation.

We attempted to test ATP production rates from glycolysis and oxidative phosphorylation in muscle fibers isolated from LPS-treated and C26 tumor mice using the Seahorse Extracellular Flux Analyzer (Schuh et al, 2012). However, the muscle fibers did not survive long enough to assess their means of energy production by this technique. We then decided to assess ATP production in IFNγ/TNF-α-treated C2C12 myotubes that are amenable to Seahorse analysis (Mookerjee et al, 2017, 2018; Hall et al, 2018). We have routinely used this in vitro cell model of muscle wasting to mimic the effect of cytokines on muscle fibers observed, in vivo, during inflammation-induced muscle wasting (Di Marco et al, 2005; Di Marco et al, 2012; Ma et al, 2017; Hall et al, 2018). We began by characterizing the validity and relevance of C2C12 myotubes in relation to our in vivo models by assessing the role of iNOS after

cytokine treatment. As previously shown, IFNγ and TNF-α induce the wasting of C2C12 myotubes over 48 h (Fig 5A and B) (Cramer et al, 2018; Hall et al, 2018; Mubaid et al, 2019). The wasting of these myotubes is associated with significant increases in both the expression of iNOS protein and nitrite levels in the media as well as deregulation of the TCA cycle and anapleurotic metabolites that feed into it (Dataset EV6 and Fig 5C and D). Consistent with our observations in the in vivo models, inhibition of iNOS with GW prevented cytokine-induced atrophy of C2C12 myotubes (Fig 5A and B; Di Marco et al, 2005; Di Marco et al, 2012). Although GW does not affect iNOS protein levels in cytokine-treated myotubes, it did inhibit, in a dose-dependent manner, iNOS-induced NO production (Fig 5C). These results were confirmed using another specific iNOS inhibitor, aminoguanidine (AMG), that has not underwent clinical development (Appendix Fig S9) (Thornalley, 2003). While cytokine treatment activated AMPK, this effect was prevented when the cells were co-treated with GW or AMG (Fig 5E and F and Appendix Fig S10A). These results were furthermore confirmed by assessing the phosphorylation of ACC, a downstream marker of AMPK (Fig 5E and F). In addition, we observed a reduction in approximately 35% in cellular ATP levels in cytokine-treated myotubes, which was prevented by GW and AMG (Fig 5G and Appendix Fig S10B). One consequence of energetic stress is the inhibition of protein synthesis via the central regulator of protein translation, mammalian target of rapamycin (mTOR; Heberle et al, 2015). AMPK activation suppresses mTOR activity and reduces protein synthesis; moreover, we have previously demonstrated that mTOR is heavily suppressed due to IFNγ/TNF-α treatment (White et al, 2013; Hall et al, 2018). Here, our experiments showed that GW restored the phosphorylation of mTOR target ribosomal protein S6 kinase (S6K) and its target, ribosomal protein S6 (S6) (Fig 5H and I).

These data demonstrate that C2C12 myotubes treated with pro-cachectic cytokines recapitulate our findings on sepsis- and tumor-induced cachexia in mice. Inhibition of iNOS using GW, however, reverts these effects.

### iNOS activity disrupts the mitochondrial electron transport chain

We next investigated whether the iNOS-dependent induction of energy crisis in myotubes undergoing wasting is caused by dysfunctions in glycolysis or OXPHOS, or both. Using the Seahorse Extracellular Flux Analyzer, we assessed oxygen consumption rate (OCR) and extracellular acidification rate (ECAR), which are correlated with oxidative respiration and glycolysis, respectively (Appendix Figs S11A and B and S12A and B). We determined the theoretical basal and maximal rates of oxidative and glycolytic ATP production (J-ATP) of C2C12 myotubes treated or not with cytokine and iNOS inhibitors (Mookerjee et al, 2017, 2018). Determination of the initial J-ATP rates, when the cells are primarily consuming exogenous glucose, allows for the assessment of the basal energetic state (J-$ATP_{total}$) of the cell, as well as preferences for either glycolytic (J-$ATP_{glycolytic}$) or mitochondrial OXPHOS (J-$ATP_{oxidative}$) ATP production. The theoretical maximal J-$ATP_{oxidative}$ and J-$ATP_{glycolytic}$ rates, respectively, can be determined after FCCP and monensin treatment (Mookerjee et al, 2017, 2018). When plotted on perpendicular axes, the area defined by these maximal rates represents the bioenergetic profile, with the intersection of these two rates defining the maximal bioenergetic capacity of the cell. Our analysis revealed that while

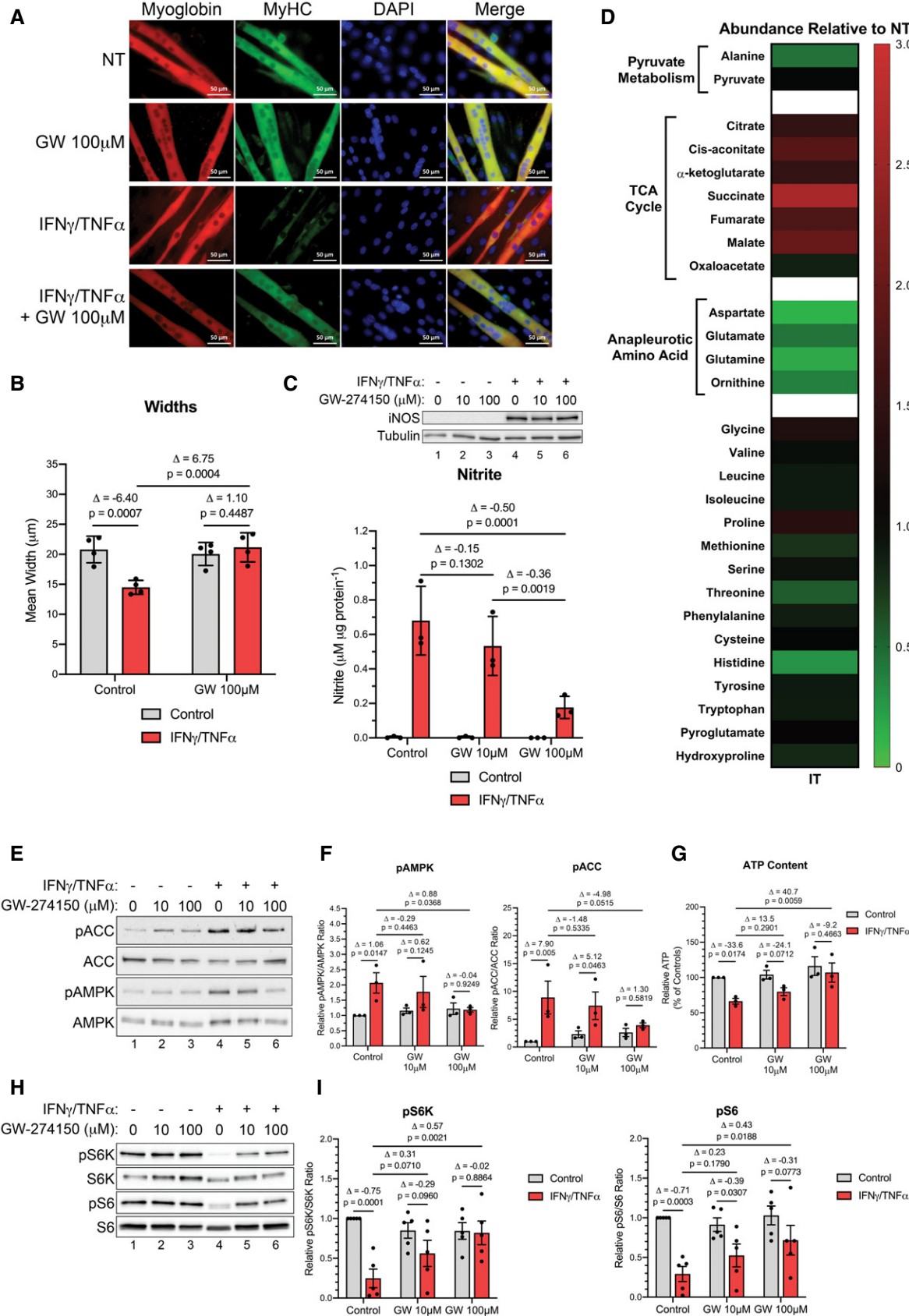

**Figure 5.**

**Figure 5.   Cytokine treatment of C2C12 myotubes alters the levels of TCA cycle intermediates and activates AMPK in an iNOS-dependent manner.**

C2C12 myotubes were treated with or without IFNγ (100 U/ml) and TNF-α (20 ng/ml) and the indicated doses of GW. Protein content and metabolites were extracted from cells 24 h after treatment and analyzed for AMPK phosphorylation, ACC phosphorylation, and total iNOS levels as well as GC-MS, respectively. Myotube integrity and widths as well as phosphorylation of S6 and S6K were assessed 48 h after treatment.

A   Representative immunofluorescence imaging for myoglobin and myosin heavy chain (MyHC) in not treated (NT) controls and IFNγ/TNF-α (IT) samples treated with or without GW. Nuclei were visualized with DAPI staining (*n* = 4).

B   Quantification of mean fiber widths (*n* = 4).

C   (*top*) Western blot analysis for iNOS and tubulin (*n* = 3). (*bottom*) Media nitrite levels (*n* = 3).

D   Heatmap visualizing mean concentration corresponding to metabolites of IFNγ/TNF-α (IT)-treated samples relative to not treated (NT) controls (*n* = 3). Red and green indicate an increase or decrease in metabolite levels, respectively.

E   Western blot analysis for pThr172-AMPK (pAMPK), total AMPK (AMPK), pSer79-ACC (pACC), and total ACC (ACC).

F   Quantification of the pAMPK-to-AMPK ratio (*left*) and the pACC-to-ACC ratio (*right*) relative to the untreated control (*n* = 3).

G   Cellular ATP content quantified as a percentage of the untreated control (*n* = 3).

H   Western blot analysis for pThr389-S6K (pS6K), total S6K (S6K), pSer235/236-S6 (pS6), and total S6 (S6) (*n* = 5).

I   Quantification of the (*left*) pS6K-to-S6K ratio and the (*right*) pS6-to-S6 ratio relative to the untreated control (*n* = 5).

Data information: Individual data points represent independent experimental replicates. Error bars represent the standard deviation (SD) of the mean. For statistical comparisons, Δ indicates the difference in mean values and *P*-values were calculated with an ANOVA followed by Fisher's LSD test.

Source data are available online for this figure.

control C2C12 myotubes have a highly oxidative metabolism, IFNγ/TNF-α caused a dramatic shift in both basal and maximal metabolic activity away from oxidative respiration toward glycolysis (Fig 6A and B and Appendix Fig S13A and B). This shift was prevented by treatment with GW or AMG in a dose-dependent manner (Fig 6A and B and Appendix Fig S13A and B). Interestingly, cytokine-treated cells had decreased basal rate of J-ATP production, indicating their glycolytic energy production was not sufficient to compensate for the loss

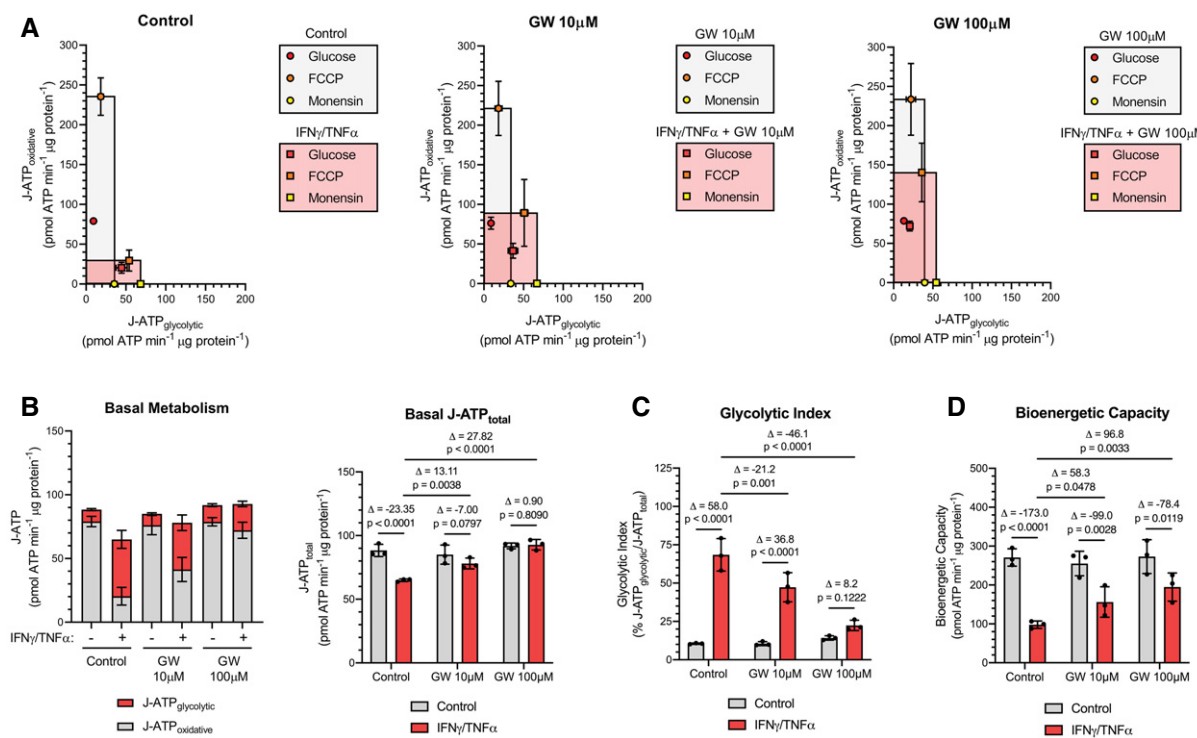

**Figure 6.   GW274150 prevents a cytokine-induced shift to aerobic glycolysis in C2C12.**

C2C12 myotubes were treated with or without IFNγ (100 U/ml) and TNF-α (20 ng/ml) and the indicated doses of GW. ATP production rates (J-ATP) from oxidative phosphorylation (oxidative) and glycolysis (glycolytic) were determined from measurements of extracellular flux 24 h after treatment.

A   Bioenergetic profiles. Highlighted squares are defined by the theoretical maximal J-ATP$_{oxidative}$ and J-ATP$_{glycolytic}$ rates.

B   (*left*) Basal J-ATP$_{glycolytic}$ and J-ATP$_{oxidative}$ rates. (*right*) Total basal J-ATP rate.

C   Glycolytic index of basal metabolism.

D   Total bioenergetic capacity.

Data information: Individual data points represent three independent experiments (*n* = 3). The data points for each experiment are calculated from the average of technical triplicates. Error bars represent the standard deviation (SD) of the mean. For statistical comparisons, Δ indicates the difference in mean values and *P*-values were calculated with an ANOVA followed by Fisher's LSD test.

of oxidative capacity (Fig 6B and Appendix Fig S13B). Importantly, J-ATP production was recovered by GW and AMG (Fig 6B and Appendix Fig S13B). These changes were reflected in the glycolytic index, a measure of how much of the total basal ATP production comes from glycolytic activity. In IFNγ/TNF-α-treated cells, the glycolytic index was 60%-95%, a dramatic increase from approximately 10% observed in control untreated cells (Fig 6C and Appendix Fig S13C). Although treatment with GW or AMG had no effects on the glycolytic index in untreated cells, it did significantly reduce it in cytokine-treated myotubes (Fig 6C and Appendix Fig S13C). Notably, the total bioenergetic capacity in cytokine-treated myotubes was significantly reduced (Fig 6D and Appendix Fig S13D). However, treatment with GW, in addition to reverting the metabolism profile back to a more oxidative state, was able to partially recover the bioenergetic capacity, allowing for restoration of the basal ATP production rate (Fig 6B and D). Although AMG was also able to restore basal ATP production, it did not have such a robust effect on bioenergetic capacity as GW (Appendix Fig S13B and D). Overall, our data indicate that inhibition of iNOS prevents the shift toward glycolytic metabolism and loss of bioenergetic capacity induced by cytokines.

To determine how iNOS impairs OXPHOS directly in muscle fibers, we used Western blot analysis to assess the integrity of the ETC complexes by determining the expression levels of the subunits of each complex that are known to be labile when the complexes are not assembled (Fig 7A and B, Appendix Fig S14A–C, and Fig EV4A–F). Cytokine treatment significantly reduced the levels of MTCO1 and SDHB, subunits of cytochrome c oxidase (COX; complex IV) and succinate dehydrogenase (SDH; complex II), by approximately 90% and 60%, respectively. The levels of these subunits were partially re-established by treatment with GW or AMG (Figs 7A and B and EV4A, C, E). This indicates that the integrity of complexes II and IV in the ETC is compromised during cytokine treatment in an iNOS-dependent manner, suggesting that the activity of these two complexes is inhibited by iNOS in myotubes undergoing wasting.

### iNOS activity disrupts the mitochondrial structure in wasting skeletal muscle

Both our *in vivo* metabolomic study and our *in vitro* analyses point to impairment of mitochondrial energy production as a common

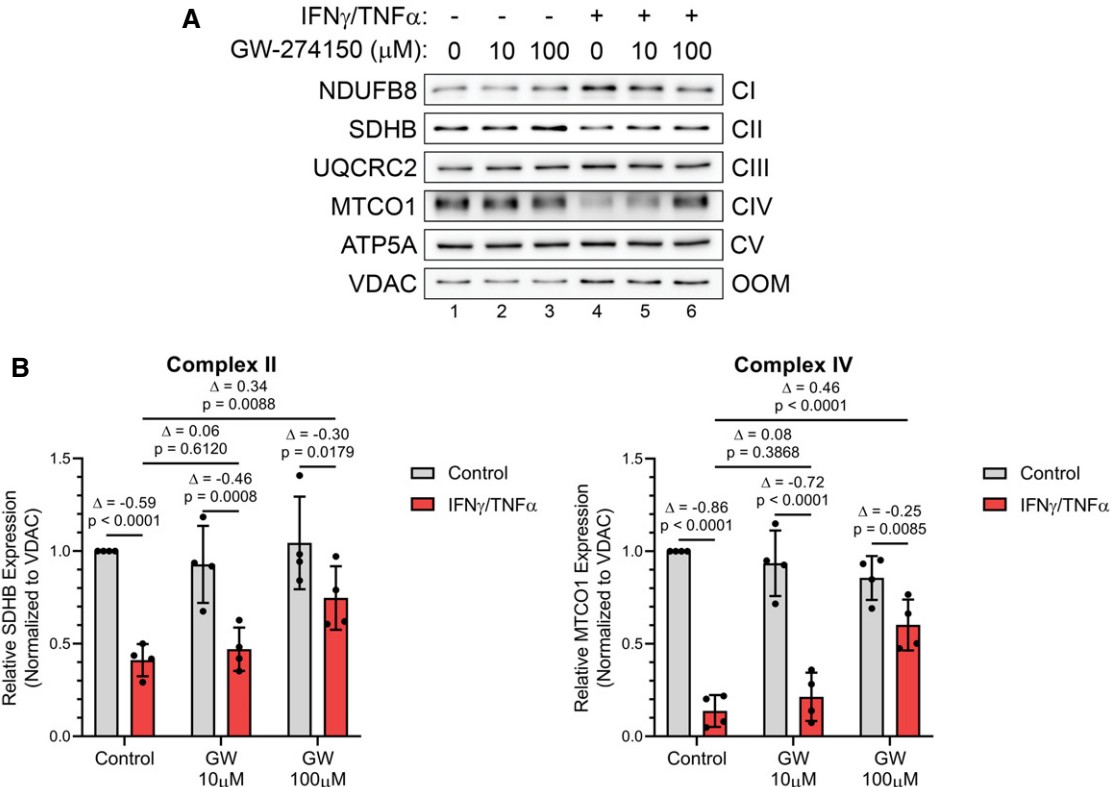

**Figure 7. Inflammation-mediated loss of complex II and IV integrity are reversed with GW274150.**

C2C12 myotubes were treated with or without IFNγ (100 U/ml) and TNF-α (20 ng/ml) and the indicated doses of GW. Protein content was extracted 24 h after treatment.
A   Western blot analysis for ETC protein complex subunits.
B   (*left*) Quantification of SDHB (complex II; CII) normalized to VDAC (outer mitochondrial membrane; OMM) and relative to untreated control. (*right*) Quantification of MTCO1 (complex IV; CIV) normalized to VDAC and relative to untreated control.

Data information: Individual data points are from four independent experimental replicates (*n* = 4). Error bars represent the standard deviation (SD) of the mean. For statistical comparisons, Δ indicates the difference in mean values and *P*-values were calculated with an ANOVA followed by Fisher's LSD test.
Source data are available online for this figure.

feature behind inflammation-induced muscle wasting. As mitochondrial morphology and number are strongly linked to its function and cellular energy production (Cogliati *et al*, 2016; Li *et al*, 2020), we assessed the impact of iNOS on these parameters in our *in vivo*

models of cachexia. Using transmission electron microscopy (TEM), we observed, in our sepsis-induced muscle wasting model, that LPS did not appear to affect the number of mitochondria in both WT and iNOS KO mice (Fig 8A left). Interestingly, however, we observed a

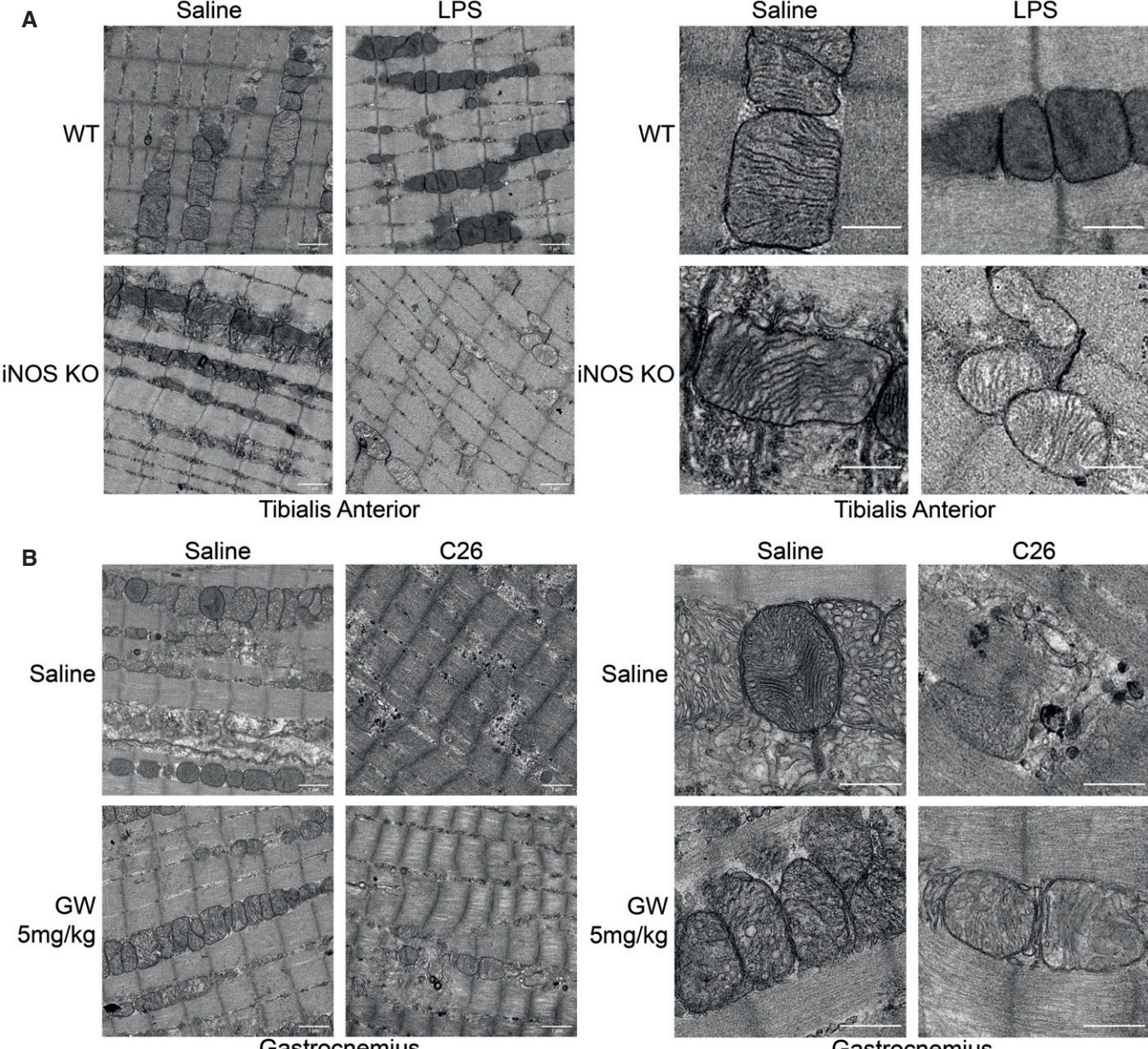

**Figure 8. Inflammation-mediated loss of mitochondrial integrity is reversed with genetic and pharmacological inhibition of iNOS.**

A   Male C57BL/6 wild-type (WT) and iNOS knockout (KO) mice were intraperitoneally injected with 1 mg kg$^{-1}$ LPS or an equivalent volume of carrier solution. Control WT, control KO, and LPS-treated KO cohorts were pair-fed (PF) to the WT LPS-treated cohorts. After 18 h, mice were euthanized, and *tibialis anterior* muscles were imaged by transmission electron microscopy. (*left*) Representative micrograph of *tibialis anterior* muscle (*n* = 2). Scale bar = 1 μm. (*right*) Zoomed section of representative image to highlight mitochondria. Scale bar = 0.5 μm.

B   Male BALB/C mice were injected subcutaneously with C26 cells (1.25 × 10$^6$ cells) or an equivalent volume of saline. After 5 days, and everyday thereafter, saline- and C26-injected mice were treated with or without GW (5 mg kg$^{-1}$). After 16 days, mice were euthanized and *gastrocnemius* muscles were imaged by transmission electron microscopy. (*left*) Representative micrograph of *gastrocnemius* muscle (*n* = 2). Scale bar = 1 μm. (*right*) Zoomed section of representative image to highlight mitochondria. Scale bar = 0.5 μm.

Data information: Images representative of 2 individual mice.

loss of cristae in the majority of LPS-treated WT mice compared with their controls and that this loss of cristae was absent in the LPS-treated iNOS KO mice (Fig 8A right). As cristae are the site of OXPHOS and significantly increase the surface area of the inner mitochondrial membrane (IMM) to house more OXPHOS complexes (Gilkerson et al, 2003; Cogliati et al, 2016; Li et al, 2020), this loss of cristae in the LPS-treated WT mice suggests that the oxidative energy production of their muscles is significantly impaired during wasting. We next analyzed mitochondrial morphology and number in our cancer-induced muscle wasting model where saline- and C26-injected mice were treated with or without GW. We observed that, when compared to control conditions, the C26 tumor environment drastically decreased the mitochondrial content in muscle fibers (Fig 8B left), causing fragmentation and loss of cristae in the mitochondria that remained (Fig 8B right). We found that treatment with GW alone did not alter the mitochondrial morphology, but tended to increase the number of mitochondria in the muscles (Fig 8B). In the context of C26 treatment, GW partially prevented the loss of mitochondria in the muscle (Fig 8B left) and maintained normal mitochondrial structure (normal cristae) (Fig 8B right). Therefore, the C26-mediated loss of mitochondrial content and disruption of structure indicates that oxidative energy production is impaired in these muscles in an iNOS-dependent manner.

Together, our models of muscle wasting demonstrate that genetic ablation of iNOS or inhibition of its activity enhances mitochondrial health and function by preventing both the loss of structure and the loss of capacity for oxidative phosphorylation by maintaining complex protein integrity.

## Discussion

In this study, we have shown that iNOS is a central downstream effector of cachexia, both in sepsis and cancer models, and that its inhibition with the clinically tested drug GW274150 can successfully ameliorate cancer-associated cachexia. In addition, the efficacy of GW confirms that iNOS is a central player in cachexia.

Notable attempts at treating cachexia have focused on ameliorating individual symptoms with little success (Speck et al, 2010; Murphy et al, 2011; Stene et al, 2013; Baldwin, 2015; Baracos et al, 2018). For example, nutritional supplementation and physical exercise to treat weight loss and loss of muscle mass demonstrated distinct beneficial outcomes but were unable to reverse cachexia (Speck et al, 2010; Murphy et al, 2011; Stene et al, 2013; Baldwin, 2015; Argiles et al, 2018). These treatments were also hindered by their dependence on patient compliance and the timing of intervention, which can limit their feasibility and effectiveness (Argilés et al, 2012; Baracos et al, 2018). Due to the lack of efficacy of the above approaches, therapeutic strategies were devised to target the inflammatory signals responsible for cachexia. Inhibition of these pro-cachectic cytokines, such as IL-6 and TNF-α, was not effective in treating cachexia and caused unwanted side effects in these patients (Wiedenmann et al, 2008; Jatoi et al, 2010; Bayliss et al, 2011). These setbacks emphasize the multifactorial nature of cachexia, where numerous pro-cachectic cytokines work together to promote the condition (Fearon et al, 2012; Baracos et al, 2018). Therefore, it is important to identify effectors of common upstream pathways that can constitute a better candidate to be targeted for

treatment. iNOS is one such effector whose importance in muscle wasting has been suggested in previous studies (Buck & Chojkier, 1996; Di Marco et al, 2005; Di Marco et al, 2012; Ma et al, 2017; Hall et al, 2018). Despite this, the mechanism driving the pro-cachectic effect of iNOS in muscles undergoing wasting has remained elusive (Buck & Chojkier, 1996; Di Marco et al, 2005; Di Marco et al, 2012; Hall et al, 2018), and the possibility of targeting iNOS to treat this deadly condition has never been explored.

Our data demonstrate that iNOS causes atrophy by impairing energy production and inducing energetic crisis in cachectic muscle which were reversed with GW or genetic ablation of iNOS (Fig 9). Indeed, this metabolic derangement is in line with previously observed cachexia-mediated impairment of oxidative metabolism and mitochondrial dysfunction in models of cachexia (Julienne et al, 2012; Puppa et al, 2012; Der-Torossian et al, 2013b; Fermoselle et al, 2013; Padrao et al, 2013; McLean et al, 2014; Brown et al, 2017; VanderVeen et al, 2018; Kunzke et al, 2019; Pin et al, 2019a; VanderVeen et al, 2019). In our cancer- and sepsis-induced cachexia models, we observed distinct alterations to metabolite profiles, which could arise from the numerous differences between the conditions of our models including the severity of the cachectic stimulus, the duration of the cachexia, and the background of the mice used. Despite this, many of the metabolic perturbations in both models were shown to be both iNOS-dependent and deleterious to the function of energy production pathways leading to energetic stress in skeletal muscle. Our data show that in muscles undergoing wasting, iNOS promoted metabolic defects, at least in part, by disrupting mitochondrial processes such as the TCA cycle by either altering metabolite levels such as α-KG and succinate or affecting anapleurotic metabolite levels such as the odd short-chain acylcarnitines. In addition, iNOS impairs acylcarnitine utilization, contributing to ineffective oxidative metabolism and lower ATP production. The alterations to mitochondrial metabolism and energy production in muscle appear to be a result of iNOS-mediated reduction in mitochondrial content and loss of cristae of the inner mitochondrial membrane that could disrupt enzymes conducting these processes as seen by the impairment of CPT1, CPT2, and OXPHOS complexes. Indeed, previous studies in inflammation-activated macrophages have shown that iNOS causes metabolic reprogramming mirroring our observations through inhibition of enzymes involved in mitochondrial processes and dysregulating mitochondrial structure (Gao et al, 2017; Bailey et al, 2019; Li et al, 2020; Palmieri et al, 2020). Therefore, we suggest that iNOS drives deleterious metabolic reprogramming in inflamed muscle to lead to energetic deficit and atrophy despite its important physiological role in activated macrophages.

Through metabolomic approaches, we identify a signature of amino acids that are altered by iNOS activity in muscle afflicted by sepsis- and cancer-induced cachexia. In our cancer model, we observed significant increases in arginine, lysine, tryptophan, and methylhistidine, which were recovered by iNOS inhibition with GW. Interestingly, the observed cancer-induced shifts in amino acid levels above are consistent with a number of recent studies (Der-Torossian et al, 2013a; QuanJun et al, 2015; Tseng et al, 2015a; Kunzke et al, 2019; Lautaoja et al, 2019b). This body of evidence raises the possibility that increases in the intramuscular concentration of these amino acids correlate with the progression of this syndrome and protein breakdown. Indeed, increases in methylhistidine and other amino acids can be markers of the muscle protein

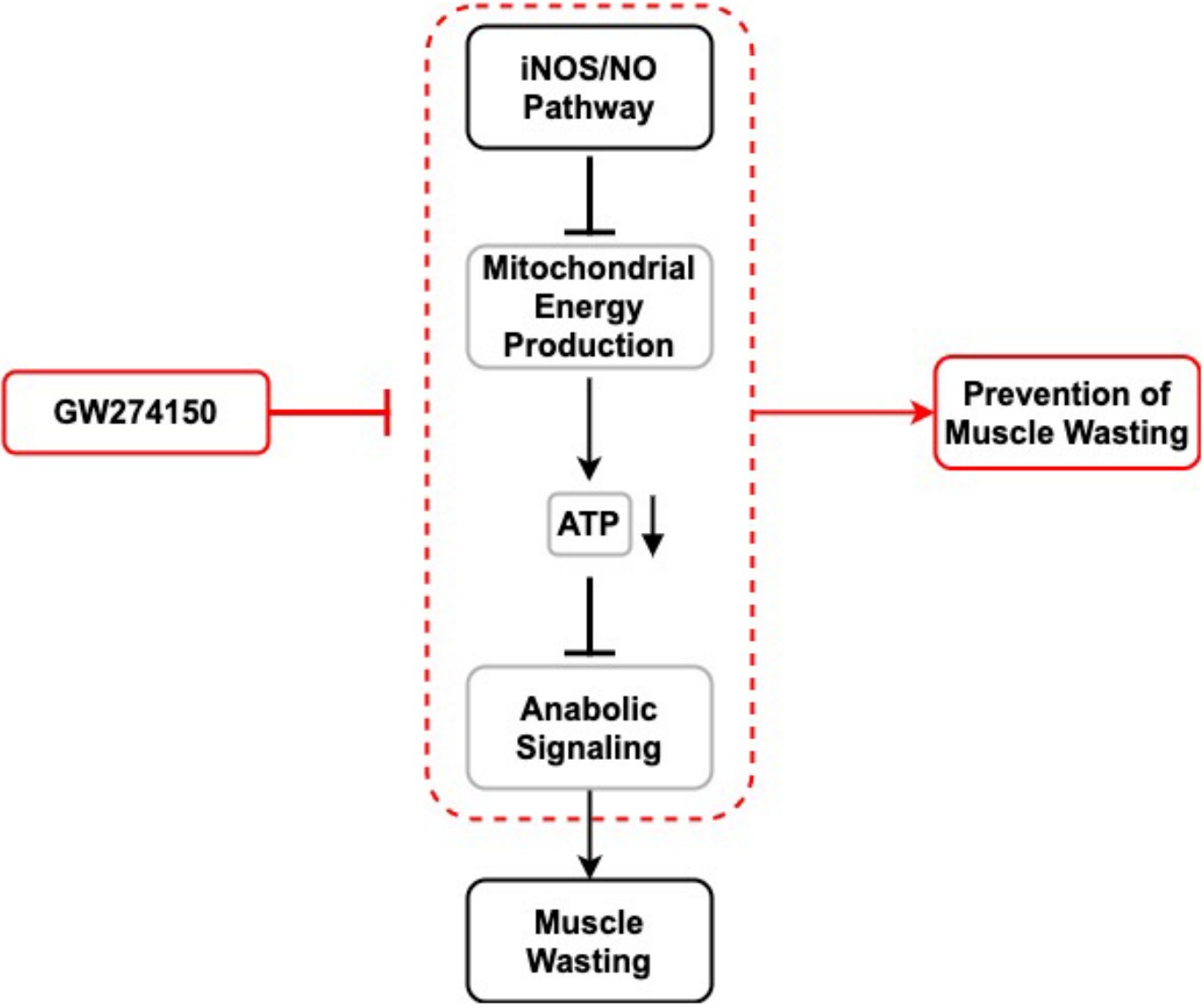

**Figure 9.  Schematic depicting the role of the iNOS/NO pathway in promoting energy crisis during cachexia-induced muscle wasting.**

*(outlined in black)* Upon activation by underlying diseases such as cancer, the iNOS/NO pathway is activated. In turn, iNOS inhibits mitochondrial energy production in skeletal muscles reducing ATP production which leads to muscle wasting. As a result of these effects, anabolic signaling is reduced and muscle loss is induced.
*(highlighted in red)* On the hand, pharmacological inhibition of iNOS activity with drugs such as GW274150 (GW) reverses these effects, preventing muscle wasting.

catabolism that contributes to wasting (QuanJun *et al*, 2015). Furthermore, in a recent study on a mouse genetic intestinal cancer model of cachexia, increases in lysine and arginine levels correlated with not only the breakdown of ETC complex proteins, but also lower overall energy levels (Kunzke *et al*, 2019). The detected increases in arginine levels in both the C26 and LPS models were of note as arginine is the substrate of iNOS in its production of NO and has been shown to promote iNOS translation (El-Gayar *et al*, 2003; Lee *et al*, 2003; Chaturvedi *et al*, 2007). The iNOS-dependent accumulation of arginine suggests that iNOS downregulates pathways that consume this metabolite as well as promotes its biosynthesis in order to facilitate iNOS expression and activity. Further investigations are needed to delineate how iNOS promotes amino acid accumulation in cachectic muscle and its consequences.

Due to the hyper-inflammatory nature of cachexia, the contribution of immune cells to muscle wasting and the overall cachectic phenotype cannot be ignored. Indeed, recent studies have demonstrated roles for cancer cachexia-induced macrophage populations in promoting muscle wasting and regulating adipose wasting (Erdem *et al*, 2019; Shukla *et al*, 2020). iNOS expression and NO secretion from inflammatory cells in the muscle micro-environment are likely contributors to muscle wasting *in vivo*. Our perturbations to iNOS, whether the full-body knockout mouse used in our septic model or our pharmacological inhibition of iNOS in our cancer model, impaired iNOS in both muscle and associated inflammatory cells. Nevertheless, we observed increases in iNOS activity visualized by 3NT levels in whole muscle lysates from cachectic mice which was associated with metabolic dysfunction in the whole

tissue and mitochondrial deformation in muscle fibers. Importantly, these effects were protected against by GW and iNOS knockout. Due to the prominent role of iNOS in immunity (Bogdan, 2001), it is also likely that iNOS will affect immune cell function in our systems. Indeed, we found that cachexia-mediated induction of IL-6 and TNF-α, two key pro-cachectic factors, was dampened by iNOS inhibition, despite not affecting alterations in macrophage levels, macrophage polarization, spleen swelling, and tumor size. These data suggest that the anti-inflammatory effects of GW and iNOS KO could, in part, contribute to the protection against cachexia and its associated metabolic defects further supporting the efficacy of iNOS impairment in treating the condition. Further studies are required to distinguish the contributions of the systemic anti-inflammatory effects of iNOS inhibition against the localized effects focusing on mitochondrial and metabolic recovery.

Increased iNOS expression has been shown in the skeletal muscle of cancer, AIDS, chronic heart failure, and COPD cachexia patients, suggesting that iNOS may be involved in the onset of cachexia under a variety of overlying conditions (Adams *et al*, 2003; Agusti *et al*, 2004; Ramamoorthy *et al*, 2009). The inhibitor used in our studies, GW274150, has been developed clinically for other inflammatory conditions, such as asthma, arthritis, and migraines (Vitecek *et al*, 2012). While GW274150 was not more effective than pre-existing treatments in reverting the pathology of these diseases, the drug was well tolerated and effective at inhibiting iNOS, with good pharmacokinetics and minimal adverse events (Vitecek *et al*, 2012). As such, drug repurposing of GW274150, or the development of other iNOS inhibitors, may be warranted for the treatment of cachexia. Specifically, the use of iNOS inhibitors may be particularly effective in cancer cachexia, given the potential pro-tumorigenic function of iNOS. Indeed, several studies have shown that iNOS inhibition can reduce tumor growth and metastasis (Kostourou *et al*, 2011; Granados-Principal *et al*, 2015; Garrido *et al*, 2017). Further clinical studies are warranted to firmly establish the therapeutic potential of iNOS inhibitors in cancer cachexia, where inhibition may play a dual role in the treatment of both cancer and muscle wasting.

# Materials and Methods

### Reagents and antibodies

IFNγ (485-MI) and TNF-α (410-MT) were purchased from R&D systems. GW274150 (HY-12119) was purchased from MedChemExpress. Aminoguanidine (396494), DAPI (10236276001), oligomycin (75351), Rotenone (R8875), antimycin A (A8674), and monensin (M5273) were purchased from Sigma-Aldrich. ECL Western Blotting Detection Reagent (RPN2106) was purchased from GE Healthcare Life Sciences.

The iNOS antibody (610431) was purchased from BD Transduction Laboratories. The tubulin antibody (DSHB Hybridoma Product 6G7; deposited by Halfter, W.M.), myosin heavy chain (DSHB Hybridoma Product MF20; deposited by Fischman, D.A.), MyHC I (DSHB Hybridoma Product BA-D5; deposited by Schiaffino, S.), and MyHC IIa (DSHB Hybridoma Product SC-71; deposited by Schiaffino, S.) antibodies were obtained from the Developmental Studies Hybridoma Bank (DSHB), created by the NICHD of the NIF and maintained at The University of Iowa, Department of Biology, Iowa City, IA 52242. The myoglobin (ab77232) and 3-NT (ab61392) antibodies as well as the Total OXPHOS Rodent Antibody Cocktail (ab110413) (containing five antibodies, one against a subunit of each ETC complex) were purchased from Abcam. VDAC (48665), pThr172-AMPKα (2535), total AMPKα (2603), pSer79-ACC (3661), total ACC (3662), pThr389-S6K (9205), total S6K (2708), pSer235/236-S6 (2211), and S6 (2317) antibodies were purchased from Cell Signaling Technology. The F4/80 conjugated to PE/Cy7 (123114), Ly6c conjugated to APC-fire750 (128046), CD206 conjugated to PE (141706), and CD86 conjugated to BV421 (105032) antibodies were purchased from BioLegend. The CD45 conjugated to BV786 (564225) antibody was purchased from BD. Laminin (L9393) antibody was purchased from Sigma-Aldrich. Horseradish peroxidase (HRP)-conjugated goat secondary antibodies against mouse (315-035-003) and rabbit (111-035-003) primary antibodies were obtained from Jackson ImmunoResearch Laboratories. Alexa Fluor™ 488-conjugated goat anti-mouse secondary (A11029), Alex Fluor™ 594-conjugated goat anti-rabbit secondary (A11072), Alexa Fluor™ 594-conjugated goat anti-mouse secondary (A11032), Alexa Fluor™ 647-conjugated goat anti-rabbit secondary (A21244), Alexa Fluor™ 488-conjugated goat anti-mouse IgG1 secondary (A21121), and Alexa Fluor™ 594-conjugated goat anti-mouse IgG2b secondary (A21145) antibodies were purchased from Thermo Fisher Scientific.

### Cell culture

C2C12 myoblast cells (ATCC, Manassas, VA, USA) were cultured as previously described (Hall *et al*, 2018). Cells were grown on 0.1% gelatin-coated culture dishes (purchased from Corning) in DMEM (Thermo Fisher 11995-065) supplemented with 20% fetal bovine serum (Sigma-Aldrich F1051) and 1% penicillin/streptomycin (Sigma-Aldrich P0781). Cells were differentiated upon reaching 90-100% confluence by switching to DMEM containing 2% horse serum (Invitrogen 16050122) and 1% penicillin/streptomycin. Myotubes were treated when visible which typically occurred three to four days after induction of differentiation. Cells were monitored for mycoplasma infection by DAPI staining.

### GC-MS metabolomic analysis

Cells were collected and lysed at the indicated time period in dry-ice chilled 80% methanol with subsequent sonication. Both the soluble and insoluble fractions were collected. Total genomic DNA (gDNA) was extracted from the insoluble fraction using phenol:chloroform: isoamyl alcohol extraction and used later to relativize metabolite levels to cellular yield. The soluble fraction was collected, and 750 ng/sample of D-myristic acid was added as an internal standard. Samples were dried down and then dissolved in 30 μl of methoxyamine hydrochloride (10 mg/ml) in pyridine and derivatized using 70 μl of N-(tert-butyldimethylsilyl)-N-methyltrifluoroacetamide (MTBSTFA) to generate tert-butyldimethylsilyl (TBDMS) esters. Metabolite levels were analyzed using an Agilent 5975C GC/MS equipped with a DB-5MS+DG (30 m × 250 μm × 0.25 μm) capillary column (Agilent J&W, Santa Clara, CA, USA). Electron impact was set at 70 eV, and a total of 1 μl of derivatized sample was injected in splitless mode with the inlet temperature set to 280°C. Helium was used as a carrier gas with a flow rate of 1.5512 ml/min (myristic acid internal standard elutes at 17.94 min). Quadrupole was set at 150°C and the interface at 285°C. The oven program was started at 60°C for 1 min,

then increased at a rate of 10°C/min until 320°C, which was held for 10 min. Data were acquired with both in scan (1–600 m/z) and selected ion (SIM) modes (McGuirk *et al*, 2013). Mass isotopomer distribution and metabolite abundance were determined using a custom algorithm developed at McGill University (McGuirk *et al*, 2013). Metabolite abundance was normalized to D-myristic acid levels and relativized to total gDNA content from the insoluble fraction.

## Media nitrite levels

Media nitrite levels were measured using GRIESS reagent as previously described (Di Marco *et al*, 2005). GRIESS reagent 1 (1% sulphanilamide, 5% phosphoric acid) and GRIESS reagent 2 (0.1% N-(1-naphthyl)-ethylenediamine dihydrochloride) were mixed in a 1:1 ratio to make GRIESS assay reagent. A standard curve was generated using a serial dilution of sodium nitrite solution. Standards and unknown media samples were mixed in a 1:1 ratio with GRIESS assay reagent and left at room temperature for 5 min. Absorbance was subsequently measured at a wavelength of 543 nm, and the nitrite concentration of unknown samples was determined by comparison with the standard curve.

## Immunoblotting

Protein content was extracted from cells using a detergent lysis buffer (50 mM HEPES pH 7.0, 150 mM NaCl, 10% glycerol, 1% Triton® X-100, 10 mM sodium pyrophosphate, 100 mM NaF, 1 mM EGTA, 1.5 mM MgCl$_2$). Protein content from muscle was extracted by homogenizing muscle in ice-cold muscle protein extraction buffer (PBS supplemented with 1% NP-40, 0.5% sodium deoxycholate, 50 mM NaF, 5 mM Na$_4$P$_2$O$_7$, and 0.1% SDS). The soluble protein fraction was clarified by centrifugation at 12,000 $g$ for 5 min at 4°C and diluted with Laemmli loading dye. Western blot analysis was performed with the Bio-Rad system. Proteins were resolved on acrylamide gels and transferred to nitrocellulose membranes using the Trans-Blot® Turbo™ system according to the manufacturer's instructions. Successful transfer and gel loading were confirmed by reversible Ponceau S staining or Stain-Free imaging. Membranes were blocked in 10% skim milk and washed three times in Tris-buffered saline containing 0.1% Tween (TBS-T). Membranes were incubated in primary antibodies diluted in either 5% skim milk or 5% BSA containing TBS-T overnight. Total OXPHOS Rodent antibody cocktail was diluted 1:1,000. VDAC was diluted 1:5,000. 3NT was diluted 1:2,000. All other antibodies were diluted as previously reported (Hall *et al*, 2018). Primary antibodies were removed by washing three times in TBS-T. Membranes were then incubated in the appropriate HRP-conjugated secondary antibodies diluted 1:5,000–1:10,000 in 5% skim milk, TBS-T for 1 h at room temperature. Membranes were washed three times in TBS-T and exposed using ECL reagent. Chemiluminescent signal was detected using either photosensitive film or a Bio-Rad ChemiDoc™ imaging system. Densitometry quantification was performed using either ImageJ software (Schneider *et al*, 2012) or Bio-Rad Image Lab™ software.

## RT–qPCR

RT–qPCR was performed as previously described (Hall *et al*, 2018). Total RNA was extracted using TRIzol® (Thermo Fisher 15596018)

according to the manufacturer's instructions. RNA quality and quantity were determined using a Thermo Fisher NanoDrop™ reader (ND-1000) and agarose gel electrophoresis. Following reverse transcription, cDNA was analyzed by qPCR with SsoFast™ Evagreen® Supermix (Bio-Rad 1725200) using primers for *F4/80* (F: 5'-GCA TCA TGG CAT ACC TGT TC-3' and 5'-GAG CTA AGG TCA GTC TTC CT-3') and *Gapdh* (F: 5'-AAG GTC ATC CCA GAG CTG AA-3' R: 5'-AGG AGA CAA CCT GGT CCT CA-3'). All levels were normalized to GAPDH and subsequently relativized to the non-treated controls.

## Extracellular flux and bioenergetics analysis

OCR and ECAR were determined using an Agilent Seahorse XFe24 Analyzer. Cells were grown and differentiated in XFe24 culture plates (1007777-004) as previously described (Hall *et al*, 2018). One hour before assessment, cells were carefully switched to XF Base Medium (103575-100) supplemented with 10 mM D-glucose (Sigma G7528) and cultured in a CO$_2$-free incubator at 37°C, as described by the manufacturer. A XFe24 sensor cartridge (102340-100), calibrated overnight in a CO$_2$-free 37°C incubator according to the manufacturer's instructions, was loaded with Oligomycin (final concentration 1 μM), FCCP (final concentration 1.5 μM), Rotenone (final concentration 1 μM), Antimycin A (final concentration 1 μM), and monensin (final concentration 20 μM). After calibration, basal extracellular flux was measured in three cycles. Drugs were subsequently injected in the indicated order, with two measurement cycles between each injection. Measurement cycles consisted of a 3-minute mix, a 2-min wait, and a 3-min measurement. Media buffering capacity was determined at the time of the experiment using a sequential injection of HCl in two wells and ranged from 0.061 to 0.065 mpH pmol H$^{+}$ $^{-1}$. Following completion of the run, individual well protein content was determined as previously described (Mookerjee *et al*, 2017). Cells were carefully washed three times in room temperature albumin-free Krebs-Ringer phosphate HEPES (KRPH) medium (2 mM HEPES, 136 mM NaCl, 2 mM NaH$_2$PO$_4$, 3.7 mM KCl, 1 mM MgCl$_2$, 1.5 mM CaCl$_2$, pH 7.4). Cells were lysed by adding 25 μl of RIPA lysis buffer (150 mM NaCl, 50 mM Tris, 1 mM EGTA, 1 mM EDTA, 1% Triton® X-100, 0.5% sodium deoxycholate, 0.1% SDS, pH 7.4) to each well. Plates were incubated on ice for 30 min and then agitated on a plate shaker for 5 min. Plates were placed back on ice, and well protein content was determined using a BCA assay (Thermo Fisher 23225) according to the manufacturer's instructions. Extracellular flux rates were normalized to protein content. Bioenergetic profiling was determined as detailed by Mookerjee *et al* (2017, 2018).

## ATP content

Cellular ATP content was determined using the Invitrogen ATP Determination Kit (A22066) as described by the manufacturer. To prepare cell lysates, media was removed from a few wells at a time by manual pipetting to avoid drying. Wells were quickly washed once with room temperature PBS, and then, cells were lysed in ATP Assay Lysis Buffer (25 mM Tris–HCl, 2 mM DTT, 2 mM EDTA, 10% glycerol, 1% Triton® X-100). After lysis buffer was added to all wells, the plate was agitated on a plate shaker for 5 min at room temperature. Plate was then placed on ice, and ATP content was determined.

## Animal models

Animal experiments were carried out with approval from the McGill University Faculty of Medicine Animal Care Committee and are in accordance with the guidelines set by the Canadian Council of Animal Care. Mice were housed in a controlled environment in sterile cages with corn-cob bedding set on a 12-h light–12-h dark cycle. Mice were provided commercial laboratory food (Harlan #2018; 18% protein rodent diet; Madison, WI) and had free access to water.

The C26 model of cancer cachexia was performed as previously described (Hall *et al*, 2018). Male BALB/C age 6–8 weeks were obtained from Jackson Laboratory. Mice were randomly assigned by cage to each treatment group. C26 adenocarcinoma cells (kindly provided by Dr. Denis Guttridge) were thawed approximately 5 days before the planned injection day. Cells were cultured in DMEM supplemented with 10% fetal bovine serum and 1% penicillin/streptomycin. Cells were kept below 70% confluency and passed a minimal number of times before injection (typically 1–2 times). C26 was prepared in chilled, sterile PBS at a concentration of $1.25 \times 10^7$ cells $ml^{-1}$. Cells were briefly warmed before injection, and 100 μl was injected subcutaneously into the right flank ($1.25 \times 10^6$ cells per mouse). For saline controls, an equivalent volume of saline was injected. Tumor growth was monitored by manual measurement with calipers. GW274150 was prepared ahead of time to a final concentration of 0.5 mg $ml^{-1}$ in PBS under sterile conditions and stored in single-use aliquots at $-20°C$. Five days after C26 injection and every day thereafter, mice were intraperitoneally injected with 10 ml $kg^{-1}$ of either GW274150 (final dose: 5 mg $kg^{-1}$) or PBS. On day 14–19 post-C26 injection, mice were anesthetized with isoflurane gas and exsanguinated by cardiac puncture. Following cervical dislocation, tissues were rapidly dissected, weighed, and snap-frozen in liquid nitrogen.

The LPS model of septic cachexia was performed as previously described (Hall *et al*, 2018). Male C57BL/6 wild-type and whole-body iNOS knockout mice on a C57BL/6 background age 8–12 weeks were obtained from Jackson Laboratories. GW-injected mice were intraperitoneally injected with 5 mg/kg dose of GW 24 h before LPS injection and then simultaneously as LPS injection. LPS was prepared fresh in a solution of 0.5% BSA dissolved in sterile PBS to a concentration of 0.1 mg $ml^{-1}$. Mice were then intraperitoneally injected with 10 ml $kg^{-1}$ to a final concentration of 1 mg $kg^{-1}$ at the beginning of the dark cycle (18:30 h to 19:30 h). Before injection, mice were separated into individual cages to allow for pair-feeding. Wild-type, LPS-injected mice were fed *ad libitum,* and all other cohorts were fed the average amount consumed by LPS-injected mice from previous studies. Mice were euthanized approximately 18 h after injection, and tissue was collected as described above.

## Grip strength

Forelimb grip strength was measured using a using a DFE II Series Digital Force Gauge (Ametek DFE II 2-LBF 10-N). Mice were allowed to acclimate to the machine the morning of C26 or LPS injection. Forelimb grip force was then measured by suspending the mouse by the tip of the tail and allowing them to establish a grip. The mouse was then gently pulled parallel to the meter until their grip was broken. Peak force was recorded, and the mouse was returned to their home cage for at least one minute. Four measurements were taken, and the maximum grip strength was determined. This process was repeated the morning of euthanasia, and the percent change in maximal grip strength was determined.

## Fluorescence-activated cell sorting (FACS)

Spleens from saline or LPS-treated WT and iNOS KO mice were removed and passed through a strainer to get single cells that were kept on ice. Residual red blood cells were lysed using ACK buffer for 3 min at room temperature. Viability was assessed using fixable viability dye eFluorYM506 (65-0866-14) (Thermo Fisher/ eBioscience). Viable cells were then analyzed using the CD45, F4/80, Ly6c, CD206, and CD86 antibodies. BD Cytofix/CytopermTM (BD Bioscience, # 554722) was used to fix the myeloid compartment according to the manufacturer's protocol. The CD86$^+$ cells or CD206$^+$ cells were gated on viable cells Cd45$^+$, F4/80, and Ly6c$^-$. Flow cytometry was done in the BD LSR Fortessa 4 lasers (405/488/561/633 nm) flow cytometer, and the results were analyzed using Flowjo_v 10.7.1_CL (Treestar).

## Muscle freezing and sectioning

Gastrocnemius and tibialis anterior muscles were mounted on 7% tragacanth gum and snap-frozen in liquid nitrogen-cooled isopentane. Samples were stored at $-80°C$ before cryosectioning. Sections (10 μm).

## Histological analysis of muscle cross-sectional area

Sections of muscles were stained using the hematoxylin and eosin (H&E) staining method. Stained samples were subsequently imaged by light microscopy. Muscle minimum Feret's diameter and cross-sectional areas were determined using ImageJ software to manually trace the circumference of individual fibers (Schneider *et al*, 2012).

## Immunofluorescence

Immunofluorescence on C2C12 myotubes was performed as previously described (Hall *et al*, 2018). In brief, cells were fixed in 3% paraformaldehyde for 30 min, permeabilized, and blocked in 0.1% Triton® X-100, 1% goat serum (Thermo Fisher 16210-064), incubated in primary antibodies against myosin heavy chain (MF-20; 1:1,000) and myoglobin (1:500) for 1 h and, after washing, Alexa Fluor® secondary antibodies (1:500) for one hour, stained with DAPI, and mounted on coverslips with VECTASHIELD® Antifade Mounting Medium (Vector Laboratories H-1000). Samples were then blinded to the experimenter, and cells were imaged with a Zeiss Observer Z1 microscope and AxioCam MRm digital camera. Myotube diameters were measured at three points along each cell using the Carl Zeiss Zen2 (blue) software.

Immunofluorescence on sections of muscles to determine fiber type composition was performed as follows: Frozen sections were equilibrated to room temperature for 20 min. Following this, the sections were fixed in 3% paraformaldehyde for 30 min, blocked in 5% goat serum for 1 h, incubated in primary antibodies against MyHC I (1:50), MyHCIIa (1:500), and laminin (1:250) overnight, and,

after washing, incubated with Alexa Fluor® secondary antibodies (1:500) for one hour. The sections were then mounted on coverslips with VECTASHIELD® HardSet Antifade Mounting (Vector Laboratories H-1400). The entirety of the sections was then imaged with a Zeiss Observer Z1 microscope and AxioCam MRm digital camera. The images were subsequently assembled together using ImageJ software, and each fiber was then classified as type I or type IIa positive. Unstained fibers with no signal were classified as being type IIb/x.

## Transmission electron microscopy

Tibialis anterior or gastrocnemius muscles were imaged at the Facility for Electron Microscopy Research at McGill University (Montreal, Quebec, Canada). The tissue samples were fixed overnight in 2.5% glutaraldehyde in 0.1 M sodium cacodylate buffer. The tissues were then postfixed with 1% $OsO_4$ + 1.5% potassium ferrocyanide in 0.1 M sodium cacodylate buffer for 2 h at 4°C and washed three times with Milli-Q water. The tissues were dehydrated in a graded series of acetone-$dH_2O$ up to 100%. Following dehydration, the tissues were infiltrated with a graded series of Epon-acetone (1:1, 2:1, 3:1), embedded in 100% Epon, and polymerized at 65°C for 48 h. Ultrathin serial sections (90–100 nm thick) were prepared from the embedded tissue blocks with a Diatome diamond knife using a Leica Microsystems EM UC6 ultramicrotome, transferred onto 200-mesh copper TEM grids, and poststained with 4% uranyl acetate for six minutes and Reynold's lead citrate for five minutes. The ultrathin sections were imaged by an FEI Tecnai G2 Spirit 120 kV TEM equipped with a Gatan UltraScan 4000 CCD Camera Model 895. The proprietary Gatan Digital Micrograph 16-bit images (DM3) were subsequently converted to unsigned 8-bit TIFF images.

## LC-MS metabolomic analysis

Tibialis anterior muscles were analyzed using the Metabolomics Innovation Centre (TMIC) Prime Metabolomics Profiling Assay. Analysis of 119 metabolites including biogenic amines, amino acids, acylcarnitines, glycerophospholipids, and organic acids in mice muscle samples was conducted by LC/DI-MS/MS developed by TMIC at University of Alberta (Edmonton, Alberta, Canada). For global assessments, the raw metabolite concentration data were analyzed using the MetaboAnalyst 4.0. The data were mean-centered and range scaled before further analysis, which included PLS-DA modeling of the data and pathway analysis using MetaboAnalyst 4.0.

## Cytokine protein analysis in sera

IL-1α, IL-1β, IL-6, and TNF-α levels in serum collected from mice were measured using BioPlex 200 Mouse Cytokine Array by Eve Technologies Corp. (Calgary, Alberta, Canada).

## Statistics and data processing

For *in vitro* studies, *n* indicates experimental replicates. For animal studies, *n* indicates the number of animals per treatment cohort. For *in vitro* studies, experimental replicates were excluded if the negative (non-treated) and positive (cytokine-treated) samples did not show an appropriate inflammatory response. For *in vivo* studies, mice were excluded if they developed complications unrelated to

### The paper explained

**Problem**
Cachexia is a syndrome that accompanies several overlying diseases, including cancer, sepsis, and AIDS, all of which promote severe, systemic inflammation. It is debilitating to patients and marked by uncontrolled and severe muscle loss, leading to a dramatic decrease in quality of life and an increase in mortality. Cachectic patients are also significantly less responsive to their treatments, thereby increasing hospitalization duration for these patients. Unsuccessful attempts at treating cachexia have focused on individually targeting each of the many inflammatory factors that cause the condition. Therefore, identifying and targeting downstream effectors is needed to address these ineffective strategies' setbacks.

**Results**
Here, we identify one such effector, named inducible nitric oxide synthase (iNOS), as a druggable molecular target to treat muscle wasting. Previous works by our group and others have highlighted the importance of iNOS as a promoter of cachexia. Here, we found that iNOS induces cachexia-induced muscle wasting by promoting mitochondrial dysfunction and metabolic crisis in skeletal muscle. Importantly, we demonstrate that inhibition of iNOS function prevents muscle wasting in sepsis- and cancer-induced models of cachexia using the clinically tested iNOS inhibitor GW274150.

**Impact**
This study highlights the feasibility of using iNOS inhibitors to treat cachexia-induced muscle wasting. The repurposing of the iNOS inhibitor GW274150 to treat muscle wasting in the clinic is now a promising prospect in alleviating this previously untreatable condition.

the cachexia phenotype or if they developed humane-intervention end-point complications early in the course of the study (e.g., ulceration of the tumor mass). Samples for measuring myotube widths and muscle fiber cross-sectional areas were blinded before acquisition of images and during quantification. Bar graphs represent the mean, with error bars showing either the standard deviation of the mean for biological replicates or the standard deviation for technical replicates. Significance *P*-values between means were computed using either using one-way ANOVA with Fisher's LSD test for multiple treatments groups, Student's *t*-test for comparison of two groups, or Kolmogorov–Smirnov test for comparison of frequency distributions in GraphPad Prism version 7 or 8, GraphPad Software, La Jolla California USA, www.graphpad. The pathway analysis of the metabolomics data set was conducted using MetaboAnalyst 4.0, www.metaboanalyst.ca. *P*-values of significantly altered pathways were determined using GlobalTest on MetaboAnalyst 4.0. P-values for the main figures, expanded view figures, and appendix figures can be found in Appendix Tables S1–S19.

## Data availability

This study includes no data deposited in external repositories.

**Expanded View** for this article is available online.

## Acknowledgements

We are thankful to Shawn McGuirk for his help in understanding the intricacies of bioenergetic profiling. This work was funded by a CIHR operating grant

(MOP-142399) and a CIHR project grant (PJT-159618) to IEG. DTH was funded by and a recipient of the James O. & Maria Meadows Studentship, the Maysie MacSporran Award, and the Charlotte and Leo Karassik Foundation Oncology PhD Fellowship. JS was funded by CIHR Studentship Award (GSD-164154).

## Author contributions

JS and DTH were responsible for experimental design and conducted all experimental investigations, with assistance from other co-authors. JS and DTH performed the formal analysis and visualization of experimental findings and prepared the original draft of the manuscript. BC assisted with sample collection and processing for Western blot analyses for *in vitro* studies and assisted with the animal studies. AO conducted the RT–qPCR analyses and the immunofluorescence staining of muscle sections, as well as assisting with the animal studies and data analysis. A-MKT assisted with animal studies performing collection and preparation of mouse samples analyzed. VSG assisted with the flow cytometry sample processing and data analysis. WM assisted with the interpretation of the flow cytometry data. SDM assisted with experimental design and data analysis. MEB assisted with data interpretation and preparation of the original manuscript. I-EG supervised experimental design, execution, data interpretation, and helped write and revise the manuscript.

## Conflict of interest

The authors declare that they have no conflict of interest.

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
