## [Review Process File · EMBO Molecular Medicine]

Pharmacological or genetic inhibition of iNOS prevents cachexia-mediated muscle wasting

Jason Sadek, Derek Hall, Bianca Colalillo, Amr Omer, Anne-Marie Tremblay, Virginie Sanguin-Gendreau, William Muller, Sergio Di Marco, Marco Bianchi, and Imed Gallouzi

DOI: [10.15252/emmm.202013591](https://doi.org/10.15252/emmm.202013591)

Corresponding author: Imed Gallouzi (imed.gallouzi@mcgill.ca)

Review Timeline:

Submission Date:	14th Oct 20
Editorial Decision:	18th Nov 20
Revision Received:	6th Apr 21
Editorial Decision:	30th Apr 21
Revision Received:	12th May 21
Accepted:	18th May 21

Editor: Lise Roth

Transaction Report:

18th Nov 2020

Dear Dr. Gallouzi,

Thank you for the submission of your manuscript to EMBO Molecular Medicine. We have now received feedback from the three reviewers who agreed to evaluate your manuscript. As you will see from the reports below, the referees acknowledge the interest of the study and are overall supporting publication of your work pending appropriate revisions.

Addressing the reviewers' concerns in full will be necessary for further considering the manuscript in our journal, and acceptance of the manuscript will entail a second round of review. EMBO Molecular Medicine encourages a single round of revision only and therefore, acceptance or rejection of the manuscript will depend on the completeness of your responses included in the next, final version of the manuscript. For this reason, and to save you from any frustrations in the end, I would strongly advise against returning an incomplete revision.

When submitting your revised manuscript, please carefully review the instructions that follow below. Failure to include requested items will delay the evaluation of your revision:

- 1) A .docx formatted version of the manuscript text (including legends for main figures, EV figures and tables). Please make sure that the changes are highlighted to be clearly visible.
- 2) Individual production quality figure files as .eps, .tif, .jpg (one file per figure).
- 3) A .docx formatted letter INCLUDING the reviewers' reports and your detailed point-by-point responses to their comments. As part of the EMBO Press transparent editorial process, the point-by-point response is part of the Review Process File (RPF), which will be published alongside your paper.
- 4) A complete author checklist, which you can download from our author guidelines (<https://www.embopress.org/page/journal/17574684/authorguide#submissionofrevisions>). Please insert information in the checklist that is also reflected in the manuscript. The completed author checklist will also be part of the RPF.
- 5) Before submitting your revision, primary datasets produced in this study need to be deposited in an appropriate public database (see <https://www.embopress.org/page/journal/17574684/authorguide#dataavailability>). Please remember to provide a reviewer password if the datasets are not yet public. The accession numbers and database should be listed in a formal "Data Availability" section (placed after Materials & Method). Please note that the Data Availability Section is restricted to new primary data that are part of this study.

6) We would also encourage you to include the source data for figure panels that show essential data. Numerical data should be provided as individual .xls or .csv files (including a tab describing the data). For blots or microscopy, uncropped images should be submitted (using a zip archive if multiple images need to be supplied for one panel). Additional information on source data and instruction on how to label the files are available at

7) Our journal encourages inclusion of *data citations in the reference list* to directly cite datasets that were re-used and obtained from public databases. Data citations in the article text are distinct from normal bibliographical citations and should directly link to the database records from which the data can be accessed. In the main text, data citations are formatted as follows: "Data ref: Smith et al, 2001" or "Data ref: NCBI Sequence Read Archive PRJNA342805, 2017". In the Reference list, data citations must be labeled with "[DATASET]". A data reference must provide the database name, accession number/identifiers and a resolvable link to the landing page from which the data can be accessed at the end of the reference. Further instructions are available at .

8) We replaced Supplementary Information with Expanded View (EV) Figures and Tables that are collapsible/expandable online. A maximum of 5 EV Figures can be typeset. EV Figures should be cited as 'Figure EV1, Figure EV2" etc... in the text and their respective legends should be included in the main text after the legends of regular figures.

- Additional Tables/Datasets should be labeled and referred to as Table EV1, Dataset EV1, etc. Legends have to be provided in a separate tab in case of .xls files. Alternatively, the legend can be supplied as a separate text file (README) and zipped together with the Table/Dataset file. See detailed instructions here:

9) The paper explained: EMBO Molecular Medicine articles are accompanied by a summary of the articles to emphasize the major findings in the paper and their medical implications for the non-specialist reader. Please provide a draft summary of your article highlighting

10) For more information: There is space at the end of each article to list relevant web links for further consultation by our readers. Could you identify some relevant ones and provide such information as well? Some examples are patient associations, relevant databases, OMIM/proteins/genes links, author's websites, etc...

11) Every published paper now includes a 'Synopsis' to further enhance discoverability. Synopses are displayed on the journal webpage and are freely accessible to all readers. They include a short stand first (maximum of 300 characters, including space) as well as 2-5 one-sentences bullet points

that summarizes the paper. Please write the bullet points to summarize the key NEW findings. They should be designed to be complementary to the abstract - i.e. not repeat the same text. We encourage inclusion of key acronyms and quantitative information (maximum of 30 words / bullet point). Please use the passive voice. Please attach these in a separate file or send them by email, we will incorporate them accordingly.

Please also suggest a striking image or visual abstract to illustrate your article. If you do please provide a png file 550 px-wide x 400-px high.

12) As part of the EMBO Publications transparent editorial process initiative (see our Editorial at <http://embomolmed.embopress.org/content/2/9/329>), EMBO Molecular Medicine will publish online a Review Process File (RPF) to accompany accepted manuscripts.

In the event of acceptance, this file will be published in conjunction with your paper and will include the anonymous referee reports, your point-by-point response and all pertinent correspondence relating to the manuscript. Let us know whether you agree with the publication of the RPF and as here, if you want to remove or not any figures from it prior to publication.

I look forward to receiving your revised manuscript.

Yours sincerely,

Lise Roth

Lise Roth, PhD
Editor
EMBO Molecular Medicine

To submit your manuscript, please follow this link:

Link Not Available

Photos 400-800 DPI

*Additional important information regarding figures and illustrations can be found at <https://bit.ly/EMBOPressFigurePreparationGuideline>

***** Reviewer's comments *****

Referee #1 (Comments on Novelty/Model System for Author):

N/A

Referee #1 (Remarks for Author):

Sadek et al present an investigation into the role of inducible nitric oxide synthase (iNOS) in muscle atrophy. The manuscript combines data from a mouse model of LPS-induced atrophy, a mouse model of cancer cachexia, and cytokine-induced wasting of myotubes to conclude that activation of iNOS causes muscle atrophy by creating an "energetic crisis" in muscle. While the manuscript certainly adds to what is known about iNOS in skeletal muscle, the manuscript is not without issues that need to be addressed prior to publication.

Major concerns

The authors cite their findings of no difference in spleen weights between either iNOS knockout mice or GW-treated mice and wild-type/untreated mice as evidence that their iNOS manipulations do not alter the level of whole-body inflammation. In this reviewer's opinion, this evidence is not sufficient to make this claim. It is well-established that iNOS plays a crucial role in macrophage polarization, and with the known importance of macrophages in cancer and the increasing evidence of a role for macrophages in cancer cachexia, the authors are unable to exclude that the major mechanism through which iNOS inhibition prevents muscle loss in their in vivo models is through immune-system mediated changes, as opposed to a direct effect of iNOS in skeletal muscle cells. At a minimum, the authors could support their findings more by providing evidence to suggest iNOS inhibition specifically in skeletal muscle for their C-26 studies, which is lacking.

While the authors provide significant evidence iNOS inhibition can prevent cytokine-induced wasting of cultured muscle cells, it is unclear if the metabolic changes that the authors have identified in wasted muscles or myotubes are directly related to iNOS or are simply a reflection of the wasted state of the muscle or myotubes. Is there a situation in which prevention of pAMPK activation would NOT prevent myotube atrophy? Stated another way, is the energetic crisis/pAMPK activation/decreased beta-oxidation a passenger in muscle/myotube atrophy, or in fact a driver? Without addressing this question, the schema in Figure 8 is not fully supported.

There are a number of incidents in which text does not fully match the statistical results presented in the associated figure. Of note, the following:

- No indication of significance for an increase in glucose or decrease in pyruvate or succinate in Figure 2E.
- No indication of significant decrease in CPT2 in 2H.
- The text should more accurately reflect the partial protection against C-26-induced atrophy and muscle dysfunction in Figure 3E-H.
- It would be more accurate to say that pAMPK only tended to increase in 4C.
- Figure 4E - no indication that aspartate was decreased.
- In Figures 7A and 7B, it appears that GW treatment partially prevented the loss of Complex II and Complex IV, yet the authors state "The levels of these subunits were re-established by treatment with GW or AMG (Figures 7A-B and S13)." More precise language should be used to describe these findings.

Minor comments

In regard to Figure 1C, the authors state that their LPS treatment did not result in a loss in body mass. While there may have been no difference between groups, a 10% loss of weight is certainly not insignificant, and the text should be clarified.

The authors state "Pathway analysis showed that, in skeletal muscle undergoing wasting, GW mainly targeted the pathways that were affected by C-26 tumors (Figure 4B)." However, the way in which 4B appears to be generated is by using the most altered pathways between saline and C-26 mice, and then graphing the difference between the C-26 and C-26-GW groups, which does not actually demonstrate that GW targeted all/most/some of the pathways altered by C-26 tumor-bearing. For example, by my reading, there is no difference in "cysteine and methionine metabolism" between C-26 and C-26-GW samples...which would suggest GW didn't target this pathway, which was significantly altered between C-26 and saline mice. The authors should consider if this is the most precise way to present their data and update their text to more accurately reflect the data as presented in Figure 4B.

Referee #2 (Comments on Novelty/Model System for Author):

Technical quality is high, but can be improved in the part analyzing the in vivo effects.

Medical impact is relevant, at least for the problem of skeletal muscle loss in cachexia, although the drug proposed has no effect in fat loss.

The authors used two different in vivo models of cachexia and an in vitro one to explore some mechanisms.

Referee #2 (Remarks for Author):

The manuscript by Sadek and coauthors stems from previous works of the same group demonstrating the role of iNOS as a central downstream effector of cachexia. Increased iNOS expression has been previously shown in the skeletal muscle of cachectic patients affected by various underlying conditions. It is also increased in LPS- and C26 cells-injected mice models of cachexia-associated muscle wasting used in this work.

In this manuscript, the authors demonstrated that iNOS suppression, obtained either with genetic ablation in iNOS-KO mice or a pharmacological inhibitor, prevents the muscle loss induced by LPS or C26 cells injection. Besides, they investigated in deep the metabolic shift associated with the pro-atrophic effects of iNOS in these two models of cachexia by analyzing specific metabolites involved in cellular energy production. They found that cachexia-inducing stimuli cause the impairment of several pathways involved in energy production that are prevented by iNOS suppression.

Finally, they used iNOS inhibitors in C2C12 myotubes undergoing IFN γ /TNF α -induced atrophy to elucidate the mechanisms driving the iNOS-mediated metabolic dysregulation and energetic crisis and found that IFN γ /TNF α treatment results in the loss of complex II and IV integrity, which can be rescued by iNOS inhibitors in a dose-dependent manner.

The work is well written, extremely interesting, and with relevant potential therapeutical applications for cachexia, as the inhibitor used in the study, GW274150, underwent already some phase II clinical trials that demonstrated its effectiveness at inhibiting iNOS, with minimal side effects.

Yet, I have some doubts that hope the authors can clarify and few remarks.

In LPS-treated mice, the authors comment that they did not observe, as previously shown (Hall et al., 2018), a loss of total body mass (Figure 1C). As a matter of fact, Fig. 1C shows that saline-treated WT mice lose about 10% of their weight in 18 h, an unexpected and apparently significant effect that was not discussed. Concerning this, data in Figure S3 B are the same as in Figure 1C or came from a separate experiment that confirms a loss of TBW in saline-treated mice?

One thing that intrigues me is that GW, despite preventing muscle wasting in both sepsis and cancer cachexia, has different outcomes in grip strength assessments, as it restores physical performances in LPS-treated mice only (Fig S6 D vs. Fig. 3G). Can the authors comment on this?

In the in vivo experiments, a group of non-cachectic mice treated with GW is missing. In some measurements (e.g., Fig. 4D and G; Fig.S6 B), the treatment of cachectic mice with GW does not revert a cachectic feature yet has strong effects, suggesting that it would be worth to ascertain the impact of GW alone on skeletal muscle.

As a minor point, normalization of organs and tissues (Fig. 1, 3, and S3) should be performed on a constant parameter, such as tibial length; otherwise, since cachexia induces total body weight changes, the measures result biased.

Referee #3 (Comments on Novelty/Model System for Author):

The work identified a novel metabolic connection between mitochondria and iNOS. However the insights of this interesting link are unclear. Therefore, the present work is preliminary and need a substantial amount of experiments to explain and sustain author's claims. A revised version is welcome.

Referee #3 (Remarks for Author):

The present work analyzed the role of iNOS in muscle wasting. Initially, the authors used the iNOS knockout mice to show that they are protected from muscle loss in catabolic condition such as sepsis. Then they used an inhibitor of iNOS and found that the treatment counteracts muscle wasting in tumor bearing mice. Metabolic profiles of muscle from septic and tumor bearing mice showed an energetic imbalance that induced AMPK. To establish the insights of iNOS-mediated muscle loss the authors moved to in vitro experiments and found that mitochondrial respiration was greatly altered by inflammatory cytokines while it was partially spared by two different iNOS inhibitors. The paper identified an interesting connection between an inflammatory enzyme (iNOS), mitochondria and muscle metabolism. The data are certainly of interest for the community.

However, several conclusions are based on preliminary data that need a further set of experiments. The authors should consider the following points:

Point 1. iNOS knockout mice should be characterized in terms of fiber type (beta oxidative/glycolytic fibers) and myosin composition in basal and catabolic conditions (LPS, C26 mediated cachexia)

Point 2. Show serum level of IL6, IL1a, IL1b and TNF α in iNOS knockout and GW-treated mice both in sepsis and Cancer cachexia conditions. Both these catabolic conditions are characterized by a pro-inflammatory status that contribute to muscle loss.

Point 3. The different cells population of the spleen must be analysed with particular attention to the macrophages belonging to M1 versus M2.

Point 4. Figure 1d: please show more than one type of muscle. Soleus, Gastrocnemius, Quadriceps and EDL should be included in the analyses. The finding that there is no more weight loss by LPS when compared with pair-fed mice suggest that either there is an excessive WAT loss in KO mice or that besides TA other muscles are not spared and contribute to weight loss.

Point 5. The metabolite profiles of wild type and KO mice dramatically differ suggesting an important change in basal metabolism that may reflect a shift in fiber types. It is important to address point 1 before making assumptions. Authors should show as supplementary figure also the ratio WT Saline/KO Saline and WT LPS/KO LPS.

Point 6. Figure 1C: AMPK activity is also allosterically regulated by the AMP levels. To establish how much AMPK is activated please check the phosphorylation status of downstream target (e.g. p-ACC). Another important point is the activation of AMPK in basal condition of iNOS knockout mice. How the authors explain this effect? Are mitochondria altered in KO mice? Does the GW treatment cause the induction of AMPK in non tumor bearing mice?

Point 7. The authors claimed that within glycolysis they observed an iNOS-dependent increase of glucose and decrease of pyruvate. However, these changes are not statistically significant and therefore, can not be used to sustain an iNOS mediated effect on glycolysis. Instead, the increase of citrate with a concomitant dramatic decrease of ketoglutarate and succinate suggest a potential involvement in glutamine synthesis (a pathway strongly altered as shown in panel B).

Point 8. The authors claimed that iNOS impairs β -oxidation during cachexia. However, panel 2H show exactly the opposite, that β -oxidation is not affected by LPS or by iNOS deletion. Authors should monitor mitochondrial morphology by EM analyses in LPS or tumor bearing mice.

Point 9. Figure S4: Because D. Guttridge showed that tumor growth causes dystrophin decrease and myofiber damage, authors should show which cells express iNOS when C26 is transplanted. Are inflammatory cells present in cachectic muscles? Is iNOS expressed in myofibers? Authors must also show that GW efficiently blocked iNOS activity in vivo.

Point 10. Show serum level of IL6, IL1a, IL1b and TNF α in tumor bearing mice +/- GW (see point 2). This is important because the work of Caretti G et al Nature Communications 2017 showed that cancer trigger IL6 expression that induce AMPK-FoxO3 axis and muscle loss.

Point 11. The authors claimed that GW partially reversed the loss of grip strength. Because the difference is not significant, the assertion is incorrect. There is no difference in force drop despite a partial protection in muscle mass.

Point 12. Figure 3H: Myofiber differs in shape especially during muscle loss (become polygonal and flattened) and therefore, diameter is affected while fiber area is independent of the shape. Please show myofiber area that is a more accurate measurement in a cross section of the muscle.

Point 13. The pattern of metabolites in cancer is dramatically different from the LPS one (e.g. Glutamine is dramatically increase in C26 and mildly affected by GW, succinate and ketoglutarate are further reduced by GW while were ameliorated by iNOS deletion in LPS, Glucose is reduced in GW and unaffected in iNOS KO, tryptophan is induced in iNOS KO and reduced in GW treated.....). Therefore, whether energy imbalance is a downstream trigger of iNOS it should be a common metabolic pathway. How Authors explains such big differences? An insight would be appreciated.

Point 14. Figure 5: Show iNOS expression in IFN γ /TNF α treated and untreated myotubes.

Point 15. Figure 5D: difficult to extrapolate data. Here TCA cycle metabolites are all induced while in LPS and C26 succinate and ketoglutarate are down. Authors should identify the link between Nitrite and mitochondria dysfunction. Is due to GMPc/PKG? Is mitochondrial fragmentation and mitophagy induced? How looks like mitochondrial network or shape in treated myotubes? Indeed, Bonetto and Zimmers group has shown that OXPHOS and TCA cycle protein are dramatically downregulated in cancer cachexia (Frontiers in Physiology. 2016).

Point-by-point rebuttal to reviewers' comments

We thank the referees for their detailed review of our manuscript and for their comments and suggestions. Having addressed their comments/suggestions, we believe that the revised manuscript is significantly improved, increasing the impact of the article.

In addition to the full manuscript, we are also providing a copy where we highlight the changes made in yellow.

Referee comments provided below are shown in bold, italics font while our responses are in normal font.

Referee #1

(Remarks for Author):

Sadek et al present an investigation into the role of inducible nitric oxide synthase (iNOS) in muscle atrophy. The manuscript combines data from a mouse model of LPS-induced atrophy, a mouse model of cancer cachexia, and cytokine-induced wasting of myotubes to conclude that activation of iNOS causes muscle atrophy by creating an "energetic crisis" in muscle. While the manuscript certainly adds to what is known about iNOS in skeletal muscle, the manuscript is not without issues that need to be addressed prior to publication.

Major concerns

The authors cite their findings of no difference in spleen weights between either iNOS knockout mice or GW-treated mice and wild-type/untreated mice as evidence that their iNOS manipulations do not alter the level of whole-body inflammation. In this reviewer's opinion, this evidence is not sufficient to make this claim. It is well-established that iNOS plays a crucial role in macrophage polarization, and with the known importance of macrophages in cancer and the increasing evidence of a role for macrophages in cancer cachexia, the authors are unable to exclude that the major mechanism through which iNOS inhibition prevents muscle loss in their in vivo models is through immune-system mediated changes, as opposed to a direct effect of iNOS in skeletal muscle cells. At a minimum, the authors could support their findings more by providing evidence to suggest iNOS inhibition specifically in skeletal muscle for their C-26 studies, which is lacking.

We would like to thank the reviewer for raising this important issue. We agree that the data showing no differences in spleen mass is not sufficient to support our claim that genetic ablation or inhibition of iNOS does not affect whole-body inflammation. For this reason, we have assessed, in our revised manuscript, other parameters of inflammation in our sepsis and cancer models of cachexia including splenic macrophage polarization,

muscle macrophage content, and the serum levels of several pro-inflammatory cytokines including IL6, IL1 α , IL1 β and TNF α (Figures EV1 and EV3B-G in the revised manuscript). These results, which have been included in our revised manuscript, indicate that, as observed with spleen mass, manipulations of iNOS did not affect the polarization of splenic macrophages as well as the accumulation of macrophages in the muscle. It did, however, reduce the levels of some of these cytokines, including IL-6 and TNF α , in the sera. Although these data suggest that iNOS inhibition prevents muscle loss, in part, through immune-system mediated changes we demonstrate, in our revised manuscript, that these effects are nonetheless associated with a decrease of iNOS/NO-mediated events in skeletal muscle. In addition to the data had showing that GW prevents iNOS activity in cultured muscle fibers (Figure 5C), in this revised manuscript we provide new data demonstrating that inhibition or genetic ablation of iNOS decreases NO-mediated effects specifically in skeletal muscle as evidenced by the decreased levels of 3-nitrotyrosine (3NT) modified proteins (Figures 1B and 3B of revised manuscript). In light of these new data, we have modified the text of our manuscript to indicate that inhibiting the direct effects of NO on skeletal muscle rescues mitochondrial dysfunction and energy crisis during the onset of inflammation-induced muscle wasting due to both changes in the immune response (Figures EV1 and EV3B-G) and a reduction of iNOS/NO-mediated effects in the muscle (Figures 1B, 3B, and 5C).

While the authors provide significant evidence iNOS inhibition can prevent cytokine-induced wasting of cultured muscle cells, it is unclear if the metabolic changes that the authors have identified in wasted muscles or myotubes are directly related to iNOS or are simply a reflection of the wasted state of the muscle or myotubes. Is there a situation in which prevention of pAMPK activation would NOT prevent myotube atrophy? Stated another way, is the energetic crisis/pAMPK activation/decreased beta-oxidation a passenger in muscle/myotube atrophy, or in fact a driver? Without addressing this question, the schema in Figure 8 is not fully supported.

We thank the reviewer for this comment. The impact of AMPK on muscle atrophy remains controversial. Although numerous reports have indicated that activation of AMPK is associated with inflammation-induced muscle wasting, we have previously shown (Hall et al., EMBO Mol Med, 2018) that the function of AMPK in this process is complex. Indeed, we have demonstrated that AMPK can either play a protective role when activated exogenously early upon induction of wasting or a detrimental role when activated later through mitochondrial dysfunction and subsequent metabolic stress. Additionally, we demonstrated, in that manuscript, that inhibition of AMPK activation does not necessarily prevent cytokine induced myotube atrophy. Therefore, although the role of AMPK activation as a driver of muscle wasting remains complex, its status in this manuscript is used as a readout of metabolic crisis and mitochondrial dysfunction. As such we have, as suggested by the reviewer, amended our schema to represent our findings more accurately.

There are a number of incidents in which text does not fully match the statistical results presented in the associated figure. Of note, the following:

- ***No indication of significance for an increase in glucose or decrease in pyruvate or succinate in Figure 2E.***
- ***No indication of significant decrease in CPT2 in 2H.***
- ***The text should more accurately reflect the partial protection against C-26-induced atrophy and muscle dysfunction in Figure 3E-H.***
- ***It would be more accurate to say that pAMPK only tended to increase in 4C.***
- ***Figure 4E - no indication that aspartate was decreased.***
- ***In Figures 7A and 7B, it appears that GW treatment partially prevented the loss of Complex II and Complex IV, yet the authors state "The levels of these subunits were re-established by treatment with GW or AMG (Figures 7A-B and S13)." More precise language should be used to describe these findings.***

We thank the reviewer for these comments and have amended our text accordingly to better reflect the statistical results presented in the associated figure.

Minor comments

In regard to Figure 1C, the authors state that their LPS treatment did not result in a loss in body mass. While there may have been no difference between groups, a 10% loss of weight is certainly not insignificant, and the text should be clarified.

We agree with the reviewer and have clarified the text accordingly to better summarize the weight loss observed in both our saline and LPS treated mice.

The authors state "Pathway analysis showed that, in skeletal muscle undergoing wasting, GW mainly targeted the pathways that were affected by C-26 tumors (Figure 4B)." However, the way in which 4B appears to be generated is by using the most altered pathways between saline and C-26 mice, and then graphing the difference between the C-26 and C-26-GW groups, which does not actually demonstrate that GW targeted all/most/some of the pathways altered by C-26 tumor-bearing. For example, by my reading, there is no difference in "cysteine and methionine metabolism" between C-26 and C-26-GW samples...which would suggest GW didn't target this pathway, which was significantly altered between C-26 and saline mice. The authors should consider if this is the most precise way to present their data and update their text to more accurately reflect the data as presented in Figure 4B.

We agree with the reviewer and have updated our text to reflect our data more accurately.

Referee #2

(Comments on Novelty/Model System for Author):

Technical quality is high, but can be improved in the part analyzing the in vivo effects. Medical impact is relevant, at least for the problem of skeletal muscle loss in cachexia, although the drug proposed has no effect in fat loss. The authors used two different in vivo models of cachexia and an in vitro one to explore some mechanisms.

(Remarks for Author):

The manuscript by Sadek and coauthors stems from previous works of the same group demonstrating the role of iNOS as a central downstream effector of cachexia. Increased iNOS expression has been previously shown in the skeletal muscle of cachectic patients affected by various underlying conditions. It is also increased in LPS- and C26 cells-injected mice models of cachexia-associated muscle wasting used in this work.

In this manuscript, the authors demonstrated that iNOS suppression, obtained either with genetic ablation in iNOS-KO mice or a pharmacological inhibitor, prevents the muscle loss induced by LPS or C26 cells injection. Besides, they investigated in deep the metabolic shift associated with the pro-atrophic effects of iNOS in these two models of cachexia by analyzing specific metabolites involved in cellular energy production. They found that cachexia-inducing stimuli cause the impairment of several pathways involved in energy production that are prevented by iNOS suppression.

Finally, they used iNOS inhibitors in C2C12 myotubes undergoing IFNg/TNFa-induced atrophy to elucidate the mechanisms driving the iNOS-mediated metabolic dysregulation and energetic crisis and found that IFNg/TNFa treatment results in the loss of complex II and IV integrity, which can be rescued by iNOS inhibitors in a dose-dependent manner.

The work is well written, extremely interesting, and with relevant potential therapeutical applications for cachexia, as the inhibitor used in the study, GW274150, underwent already some phase II clinical trials that demonstrated its effectiveness at inhibiting iNOS, with minimal side effects.

Yet, I have some doubts that hope the authors can clarify and few remarks.

In LPS-treated mice, the authors comment that they did not observe, as previously shown (Hall et al., 2018), a loss of total body mass (Figure 1C). As a matter of fact, Fig. 1C shows that saline-treated WT mice lose about 10% of their weight in 18 h, an unexpected and apparently significant effect that was not discussed. Concerning this, data in Figure S3 B are the same as in Figure 1C or came from a separate experiment that confirms a loss of TBW in saline-treated mice?

We thank the reviewer for raising this point and we apologize for the lack of clarity. LPS, as previously described, is known to affect food consumption rates in mice (Hall et al., EMBO Mol Med, 2018; Braun et al., FASEB J, 2013). As such, the saline treated wild type as well as the saline and LPS treated iNOS KO mice were pair-fed to the consumption rate of the LPS-treated wild-type mice in order to account for variations which may occur due to differences in food consumption. Although we observed that pair feeding caused a 10-12% decrease in body weight in both saline treated WT and iNOS KO mice, treatment with LPS did not further affect this loss of mass (Figure EV2A of revised manuscript). Additionally, as stated in the figure legend, some of the data shown in Figure S4B is indeed the same as what was shown in Figure 1C of our original manuscript. We did so, however, to demonstrate that the effects of GW in the sepsis model is similar to those seen in our iNOS knockout mice. We amended the text to include these clarifications.

One thing that intrigues me is that GW, despite preventing muscle wasting in both sepsis and cancer cachexia, has different outcomes in grip strength assessments, as it restores physical performances in LPS-treated mice only (Fig S6 D vs. Fig. 3G). Can the authors comment on this?

We thank the reviewer for this insight. We agree that there is no significant recovery in grip strength between our C26-tumour bearing mice treated with or without GW. This, however, is likely explained by our newly included data demonstrating that prolonged GW treatment has an impact on grip strength of non-tumor bearing mice. Our results (Figure 3I of revised manuscript) nevertheless show that GW partially prevents the loss of grip strength seen in the C26-tumour bearing mice, restoring it to the same level as what was observed in mice treated with GW alone. We have amended our text accordingly to reflect these observations.

In the in vivo experiments, a group of non-cachectic mice treated with GW is missing. In some measurements (e.g., Fig. 4D and G; Fig.S6 B), the treatment of cachectic mice with GW does not revert a cachectic feature yet has strong effects, suggesting that it would be worth to ascertain the impact of GW alone on skeletal muscle.

We would like to thank the reviewer for this suggestion. We have included, in the revised manuscript, new data demonstrating that GW alone does not appear to have an effect on skeletal muscle mass, metabolome or mitochondrial function (Figures 3, 4C, S5, and S8 of revised manuscript).

As a minor point, normalization of organs and tissues (Fig. 1, 3, and S3) should be performed on a constant parameter, such as tibial length; otherwise, since cachexia induces total body weight changes, the measures result biased.

We would like to thank the reviewer for this suggestion and we apologize for the confusion. In fact, in our study, the mass of skeletal muscle and tissues was normalized to the initial body weight as previously described (Pin et al., J Cachexia Sarcopenia Muscle, 2019; Michaelis et al., J Cachexia Sarcopenia Muscle, 2017; Pin et al., FASEB J, 2019; Parajuli et al., Developmental Cell, 2018). We amended the text to clarify this point.

Referee #3

(Comments on Novelty/Model System for Author):

The work identified a novel metabolic connection between mitochondria and iNOS. However the insights of this interesting link are unclear. Therefore, the present work is preliminary and need a substantial amount of experiments to explain and sustain author's claims. A revised version is welcome.

(Remarks for Author):

The present work analyzed the role of iNOS in muscle wasting. Initially, the authors used the iNOS knockout mice to show that they are protected from muscle loss in catabolic condition such as sepsis. Then they used an inhibitor of iNOS and found that the treatment counteracts muscle wasting in tumor bearing mice. Metabolic profiles of muscle from septic and tumor bearing mice showed an energetic imbalance that induced AMPK. To establish the insights of iNOS-mediated muscle loss the authors moved to in vitro experiments and found that mitochondrial respiration was greatly altered by inflammatory cytokines while it was partially spared by two different iNOS inhibitors. The paper identified an interesting connection between an inflammatory enzyme (iNOS), mitochondria and muscle metabolism. The data are certainly of interest for the community. However, several conclusions are based on preliminary data that need a further set of experiments. The authors should consider the following points:

Point1. iNOS knockout mice should be characterized in terms of fiber type (beta oxidative/glycolytic fibers) and myosin composition in basal and catabolic conditions (LPS, C26 mediated cachexia)

We would like to thank the reviewer for this suggestion. We have included new data characterizing muscle fiber type in both normal and catabolic conditions (Figures EV2D and EV3J of revised manuscript).

Point2. Show serum level of IL6, IL1a, IL1b and TNF α in iNOS knockout and GW-treated mice both in sepsis and Cancer cachexia conditions. Both these catabolic conditions are characterized by a pro-inflammatory status that contribute to muscle loss.

We are grateful to the reviewer for this suggestion. We have included new data measuring serum levels of IL6, IL1 α , IL1 β and TNF α in both our sepsis and cancer cachexia models. Our new data (Figures EV1F-I and EV3D-G of revised manuscript) indicate that the iNOS affects the secretion of some of these cytokines in our sepsis and cancer cachexia models. The text was amended to include these new observations.

Point3. The different cells population of the spleen must be analysed with particular attention to the macrophages belonging to M1 versus M2.

We would like to thank the reviewer for this suggestion. We have included new data characterizing M1 versus M2 polarization in the spleen (EV1B-D).

Point4. Figure1d: please show more than one type of muscle. Soleus, Gastrocnemius, Quadriceps and EDL should be included in the analyses. The finding that there is no more weight loss by LPS when compared with pair-fed mice suggest that either there is an excessive WAT loss in KO mice or that besides TA other muscles are not spared and contribute to weight loss.

We would like to thank the reviewer for this suggestion. Indeed, we have included new data (Figures 1E-F, 3F-G, and EV2B of revised manuscript) showing that, in addition to the TA, the mass of the soleus and quadriceps muscles are also decreased in our sepsis model of muscle wasting.

Point5. The metabolite profiles of wild type and KO mice dramatically differ suggesting an important change in basal metabolism that may reflect a shift in fiber types. It is important to address point1 before making assumptions. Authors should show as supplementary figure also the ratio WT Saline/KO Saline and WT LPS/KO LPS.

We would like to thank the reviewer for this suggestion. As stated above, we have included new data (Figures EV2D of revised manuscript) demonstrating that the composition of fiber type in iNOS knockout mice is similar to their wild type counterparts in the presence or absence of LPS. We have amended our text to indicate that differences in metabolism is not due to a significant shifts in fiber types. Furthermore, we have added a supplementary figure (Figure S1) demonstrating the ratio of metabolites between WT Saline/KO Saline and WT LPS/KO LPS.

Point6. Figure 1C: AMPK activity is also allosterically regulated by the AMP levels. To establish how much AMPK is activated please check the phosphorylation status of downstream target (e.g. p-ACC). Another important

point is the activation of AMPK in basal condition of iNOS knockout mice. How the authors explain this effect? Are mitochondria altered in KO mice? Does the GW treatment cause the induction of AMPK in non tumor bearing mice?

We thank the reviewer for this comment. We have added new data assessing ACC phosphorylation in our C2C12 model to establish that AMPK is activated during our cachectic treatment (Figure 5E-F). Despite our best efforts we were, unfortunately, unable to obtain clear *in vivo* data on pACC levels. Regarding the basal activation of AMPK in iNOS knockout mice, although we cannot comment on the origin of this basal activation as it appears that the mitochondria of KO mice are normal and GW does not induce AMPK, we do believe that it may explain, at least in part, the protection of iNOS KO mice against muscle wasting. Indeed, in a previous study, we found that the direct activation of AMPK with drugs such as AICAR or A-769662 before the onset of cachexia-induced afforded myotubes and skeletal muscle protection against wasting. The origin of the high basal activation of AMPK in iNOS KO mice will be the subject of another study.

Point7. The authors claimed that within glycolysis they observed an iNOS-dependent increase of glucose and decrease of pyruvate. However, these changes are not statistically significant and therefore, can not be used to sustain an iNOS mediated effect on glycolysis. Instead, the increase of citrate with a concomitant dramatic decrease of ketoglutarate and succinate suggest a potential involvement in glutamine synthesis (a pathway strongly altered as shown in panel B).

We thank the reviewer for this comment and amended our text as described above in our rebuttal to comments raised by referee #1. In addition, a second assessment of glutamine concentration showed no change in the steady-state levels in our manipulations.

Point8. The authors claimed that iNOS impairs b-oxidation during cachexia. However, panel 2H show exactly the opposite, that b-oxidation is not affected by LPS or by iNOS deletion. Authors should monitor mitochondrial morphology by EM analyses in LPS or tumor bearing mice.

We are grateful to the reviewer for these suggestions. In the description of Figure 2H in our revised manuscript, we have adjusted our conclusions to reflect the specific factors affected by our treatments. We have, additionally, included new transmission electron microscopy data (Figure 8 of revised manuscript) demonstrating that iNOS affects mitochondrial content, structure, and function in LPS or tumor bearing mice.

Point9. Figure S4: Because D. Guttridge showed that tumor growth causes dystrophin decrease and myofiber damage, authors should show which cells express iNOS when C26 is transplanted. Are inflammatory cells present in

cachectic muscles? Is iNOS expressed in myofibers? Authors must also show that GW efficiently blocked iNOS activity in vivo.

We thank the reviewer for these comments. In order to address these comments, we have included new data (Figures 5C, EV1E, EV3C, 1B, and 3B of revised manuscript) demonstrating that 1) C2C12 myotubes express iNOS in response to IFN γ and TNF α , 2) iNOS does not affect macrophage content in muscle and 3) as described above in response to reviewer 1, the levels of 3-nitrotyrosine in cachectic muscles are decreased due to treatment with GW indicating that GW blocked iNOS *in vivo*.

Point10. Show serum level of IL6, IL1a, IL1b and TNFa in tumor bearing mice +/- GW (see point2). This is important because the work of Caretti G et al Nature Communications 2017 showed that cancer trigger IL6 expression that induce AMPK-FoxO3 axis and muscle loss.

We thank the reviewer for this suggestion. As stated above we have included new data measuring serum levels of IL6, IL1 α , IL1 β and TNF α levels in both our sepsis and cancer cachexia models. Notably, our new data (Figures EV1F-I and EV3D-G of revised manuscript) demonstrate that the genetic ablation of iNOS or its inhibition with GW respectively affects the secretion of IL-6 in our sepsis and cancer cachexia models.

Point11. The authors claimed that GW partially reversed the loss of grip strength. Because the difference is not significant, the assertion is incorrect. There is no difference in force drop despite a partial protection in muscle mass.

We thank the reviewer for this comment. We agree that there is no significant recovery in grip strength between our C26-tumour bearing mice treated with or without GW. As per data added in response to a comment by referee #2, we show that our GW treatment in this experiment tended to lower the grip strength of non-tumor bearing mice, explaining the lack of significance in the recovery.

Point12. Figure 3H: Myofiber differs in shape especially during muscle loss (become polygonal and flatted) and therefore, diameter is affected while fiber area is independent of the shape. Please show myofiber area that is a more accurate measurement in a cross section of the muscle.

We would like to thank the reviewer for this suggestion. We have added data (Figures EV2C and EV3I of revised manuscript) showing myofiber cross-sectional area and replaced the images in the figure to better represent the data.

Point13. The pattern of metabolites in cancer is dramatically different from the LPS one (e.g Glutamine is dramatically increase in C26 and mildly affected by GW, succinate and ketoglutarate are further reduced by GW while were ameliorated by iNOS deletion in LPS, Glucose is reduced in GW and unaffected in iNOS KO, tryptophan is induced in iNOS KO and reduced in GW treated.....).

Therefore, whether energy imbalance is a downstream trigger of iNOS it should be a common metabolic pathway. How Authors explains such big differences? An insight would be appreciated.

We thank the reviewer for this insight. We believe that the numerous differences in the conditions of our models including the background of the mice (C57BL/6 vs BALB/C), the mode and severity of cachexia induction (LPS vs C26 adenocarcinoma), and the duration of the cachexia (acute- 18h vs chronic – 16 days) caused the substantial differences in the metabolites affected by each model. A statement in the discussion section was included to clarify this point.

Point14. Figure 5: Show iNOS expression in IFN γ /TNF α treated and untreated myotubes.

We would like to thank the reviewer for this suggestion. We have added data (Figure 5C of revised manuscript) showing iNOS expression in IFN γ /TNF α myotubes

Point15. Figure5D: difficult to extrapolate data. Here TCA cycle metabolites are all induced while in LPS and C26 succinate and ketoglutarate are down. Authors should identify the link between Nitrite and mitochondria dysfunction. Is due to GMPc/PKG? Is mitochondrial fragmentation and mitophagy induced? How looks like mitochondrial network or shape in treated myotubes? Indeed, Bonetto and Zimmers group has shown that OXPHOS and TCA cycle protein are dormatically downregulated in cancer cachexia (Forntiers in Physiology. 2016).

We would like to thank the reviewer for this comment. Regarding the C2C12 metabolite data, like our statements in response to point 14, we believe that the nature of the C2C12 model vs our *in vivo* models is the cause of these distinct metabolic profiles. The C2C12 model ignores systemic effects found in the mouse models where metabolic organs such as the liver and adipose would affect muscle metabolism. We agree with the reviewer that elucidating the link between nitrite and mitochondrial dysfunction and the role of GMPc/PKG could prove to be interesting. We, however, believe that addressing this is beyond the scope of our study. Nonetheless, we have added EM data demonstrating derangement of mitochondrial morphology in cachectic conditions which was correlated with iNOS impairment, suggesting that mitochondrial dynamics are affected by iNOS function in our cachectic models.

30th Apr 2021

Dear Dr. Gallouzi,

Thank you for the submission of your revised manuscript to EMBO Molecular Medicine, and please accept my apologies for the delay in getting back to you, which is due to the current high number of manuscripts submitted to our editorial office.

We have now received the enclosed reports from the referees. As you will see, they are supportive of publication, and I am therefore pleased to inform you that we will be able to accept your manuscript, once the following minor points will be addressed:

1) Please address the minor point from referee #1.

2) Main manuscript text:

- Please answer/correct the changes suggested by our data editors in the main manuscript file (in track changes mode). This file will be sent to you in the next couple of days. Please use this file for any further modification.
- Please remove the highlights in the text (including in the appendix).
- Please provide up to 5 keywords.
- Material and methods: please indicate the housing and husbandry conditions of the mice.
- Please add a 'Data availability' section after the 'Material and Methods'. This section is meant to list the primary datasets produced in this study deposited in an appropriate public database (see <https://www.embopress.org/page/journal/17574684/authorguide#dataavailability>). Please note that the Data Availability Section is restricted to new primary data that are part of this study. If no new dataset was generated, please indicate: "This study includes no data deposited in external repositories". Please remove the 'Data and materials availability' section you currently have after the acknowledgements.
- The funding information provided in the manuscript (Acknowledgements) do not match the information provided in the submission system. Please add in the submission system James O. & Maria Meadows Studentship, the Maysie MacSporran Award, the Charlotte and Leo Karassik Foundation Oncology PhD Fellowship, and CIHR Studentship Award (GSD-164154)
- Please rename 'Competing Interests' as 'Conflicts of interest'.

3) Figures:

- Statistics: Please indicate in all main, EV and appendix figures (or in their legends) the exact p-values (including for non-significant p values, ns). You may provide these values as a supplementary table in an Appendix file.
- Please make sure that all figures/figure panels are referenced in the main text (callouts are missing for Fig. 7 panels A and B, and Fig. EV4 panels A to F).
- The figure legends for EV figures should be grouped together after the main figure legends.

4) Checklist: please fill in section F18. If no new dataset was generated, please indicate "This study includes no data deposited in external repositories"

5) Source Data: Thank you for providing Source Data for Figure 7. Please upload the source data as PDF file.

6) Thank you for providing a synopsis text. I added minor edits to fit our style and format, please let me know if you agree with the following:

Cachexia is a condition marked by severe skeletal muscle atrophy in patients affected by diseases such as cancer or sepsis. The inflammation-induced factor inducible nitric oxide synthase (iNOS) was found to promote muscle wasting by causing mitochondrial dysfunction and energetic stress.

- In an LPS-induced model of septic cachexia, muscle wasting was prevented by iNOS impairment through genetic knockout or use of the pharmacological inhibitor GW274150.
- Muscle wasting was prevented by GW274150 in the C26 adenocarcinoma-induced model of cancer cachexia
- Metabolic processes, including glycolysis, the TCA cycle, acylcarnitine metabolism, and oxidative phosphorylation were dysregulated by iNOS.
- Cachexia-induced loss of mitochondrial structure and content in skeletal muscle was prevented by iNOS inhibition.

Please also suggest a striking image or visual abstract to illustrate your article as a png, tiff or jpeg file (550 px-wide x 400-px high).

7) As part of the EMBO Publications transparent editorial process initiative (see our Editorial at <http://embomolmed.embopress.org/content/2/9/329>), EMBO Molecular Medicine will publish online a Review Process File (RPF) to accompany accepted manuscripts.

This file will be published in conjunction with your paper and will include the anonymous referee reports, your point-by-point response and all pertinent correspondence relating to the manuscript. Let us know whether you agree with the publication of the RPF.

I look forward to receiving your revised manuscript.

Yours sincerely,

Lise Roth

Lise Roth, PhD
Editor
EMBO Molecular Medicine

To submit your manuscript, please follow this link:

Link Not Available

The system will prompt you to fill in your funding and payment information. This will allow Wiley to send you a quote for the article processing charge (APC) in case of acceptance. This quote takes into account any reduction or fee waivers that you may be eligible for. Authors do not need to pay any fees before their manuscript is accepted and transferred to our publisher.

***** Reviewer's comments *****

Referee #1 (Comments on Novelty/Model System for Author):

I selected "medium" for medical impact and novelty because the limitations of the C-26 model are not insignificant, and additional work would likely be required before a clinical trial was possible for cancer cachexia. The model system is not completely inadequate - it's just that additional data from other systems would likely be required.

Referee #1 (Remarks for Author):

The authors have admirably responded to all of my concerns with their previous version of this manuscript, which is not a trivial accomplishment. The only remaining issue that I have is that the running title is currently listed as "draggable" when I believe the authors intend for it to be "druggable".

Referee #2 (Remarks for Author):

Is suitable for publication

Referee #3 (Remarks for Author):

The revised version is improved and the authors addressed all my concerns

The authors performed the requested editorial changes.

18th May 2021

Dear Dr. Gallouzi,

Thank you for sending the revised files. I have looked at everything, and all is fine. I am therefore very pleased to accept your manuscript for publication in EMBO Molecular Medicine!

Your manuscript will be sent to our publisher to be included in the next available issue of EMBO Molecular Medicine.

I have noted that you suggested a nice cover image. Covers are chosen upon discussion within the editorial team once all the articles published within one issue are defined. We will get back to you once we have reached a decision.

Congratulations on a nice study!

With kind regards,

Lise Roth

Lise Roth, Ph.D
Editor
EMBO Molecular Medicine

Corresponding Author Name: Imed Gallouzi
Journal Submitted to: EMBO Molecular Medicine
Manuscript Number: 2020-13591